# E3 ligase FBXW7 suppresses brown fat expansion and browning of white fat

Jian Yu[1,2,3,4,11], Xuejiang Gu[5,11], Yingying Guo[6,11], Mingyuan Gao[2,11], Shimiao Cheng[2], Meiyao Meng[2], Xiangdi Cui[2], Zhe Zhang[2], Wenxiu Guo[2], Dandan Yan[6], Maozheng Sheng[2], Linhui Zhai[7], Jing Ji[8], Xinhui Ma[8], Yu Li[2], Yuxiang Cao[2], Xia Wu[2], Jiejie Zhao [ID][9], Yepeng Hu[5], Minjia Tan [ID][7], Yan Lu[6,9], Lingyan Xu [ID][2,10 ✉], Bin Liu [ID][8 ✉], Cheng Hu [ID][1,6 ✉] & Xinran Ma [ID][1,2,3,4,10 ✉]

## Abstract

**Thermogenic fat, including brown and beige fat, dissipates heat via thermogenesis and enhances energy expenditure. Thus, its activation represents a therapeutic strategy to combat obesity. Here, we demonstrate that levels of F-box and WD repeat domain-containing 7 (FBXW7), an E3 ubiquitin protein ligase, negatively correlate with thermogenic fat functionality. FBXW7 overexpression in fat suppresses energy expenditure and thermogenesis, thus aggravates obesity and metabolic dysfunctions in mice. Conversely, FBXW7 depletion in fat leads to brown fat expansion and browning of white fat, and protects mice from diet induced obesity, hepatic steatosis, and hyperlipidemia. Mechanistically, FBXW7 binds to S6K1 and promotes its ubiquitination and proteasomal degradation, which in turn impacts glycolysis and brown preadipocyte proliferation via lactate. Besides, the beneficial metabolic effects of FBXW7 depletion in fat are attenuated by fat-specific knockdown of S6K1 in vivo. In summary, we provide evidence that adipose FBXW7 acts as a major regulator for thermogenic fat biology and energy homeostasis and serves as potential therapeutic target for obesity and metabolic diseases.**

**Keywords** Obesity; Thermogenic Fat; Brown Fat Expansion; Browning of White Fat; Ubiquitination
**Subject Categories** Metabolism; Post-translational Modifications & Proteolysis

## Introduction

Obesity is one of the major risk factors for various chronic diseases, including diabetes, hypertension, hyperlipidemia and hepatic steatosis (Piché et al, 2020). Obesity, caused by an imbalance of nutrient intake and energy expenditure, is characterized of excessive storage of fat in adipose tissues (Caballero, 2019). Adipose tissues have been divided into brown, beige, and white adipose tissues according to their distinct location, morphology, and function (Cinti, 2018). White adipose tissue (WAT) stores energy in the form of triglycerides (Peirce et al, 2014). Thermogenic fat, including brown (BAT) and beige adipose tissues generate heat by decoupling proton gradient from oxidative phosphorylation and releasing it through uncoupling protein (UCP1) in mitochondrial membrane. Compared to inducible beige fat, brown fat acts constitutively (Cannon and Nedergaard, 2004; Petrovic et al, 2010; Wu et al, 2012). In rodents, thermogenic capacity is retained in both brown and beige fat. Adult humans also possess brown and beige fat that exist at different depth in the suprascapular area and around neck with inducible high metabolic activity and chemical energy consumption upon cold stimulation (Bartelt and Heeren, 2014; Kajimura et al, 2015; Cypess et al, 2009). Clinical studies have demonstrated that thermogenic fat content is negatively correlated with BMI, body fat content and fasting glucose levels, suggesting activation of thermogenic fat may improve metabolic dysfunctions (Carey et al, 2013; Cypess et al, 2015; Sidossis and Kajimura, 2015).

Transcriptional activation of thermogenic fat function has been extensively studied in thermogenic adipocytes. For example, core transcription factors/cofactors such as PRDM16, PGC1α, and PPARγ have been well characterized in modulating target genes mRNA abundance (Seale et al, 2007; Seale et al, 2008). Notably, recent studies have shown that post-translational modification, particularly the ubiquitination modification and the subsequent

[1]Joint Center for Translational Medicine, Fengxian District Central Hospital, Fengxian District, Shanghai 201400, China. [2]Shanghai Key Laboratory of Regulatory Biology, Institute of Biomedical Sciences and School of Life Sciences, East China Normal University, Shanghai 200241, China. [3]Chongqing Key Laboratory of Precision Optics, Chongqing Institute of East China Normal University, Chongqing 401120, China. [4]Shanghai Frontiers Science Center of Genome Editing and Cell Therapy, Shanghai Key Laboratory of Regulatory Biology, Institute of Biomedical Sciences and School of Life Sciences, East China Normal University, Shanghai 200241, China. [5]Department of Endocrine and Metabolic Diseases, the First Affiliated Hospital of Wenzhou Medical University, Wenzhou, Zhejiang 325000, China. [6]Shanghai Diabetes Institute, Shanghai Key Laboratory of Diabetes Mellitus, Shanghai Clinical Centre for Diabetes, Shanghai 200233, China. [7]State Key Laboratory of Drug Research, Shanghai Institute of Materia Medica, Chinese Academy of Sciences, Shanghai 201203, China. [8]Jiangsu Key Laboratory of Marine Pharmaceutical Compound Screening, College of Pharmacy, Jiangsu Ocean University, Lianyungang 222005, China. [9]Ministry of Education Key Laboratory of Metabolism and Molecular Medicine, Department of Endocrinology and Metabolism, Zhongshan Hospital, Fudan University, Shanghai 200000, China. [10]Institute for Aging, East China Normal University, Shanghai 200241, China. [11]These authors contributed equally: Jian Yu, Xuejiang Gu, Yingying Guo, Mingyuan Gao. ✉E-mail: lyxu@bio.ecnu.edu.cn; liubin@jou.edu.cn; alfredhc@sjtu.edu.cn; xrma@bio.ecnu.edu.cn

ubiquitin-proteasome pathway, plays important roles in maintaining energy homeostasis in metabolic organs (Haberecht-Müller et al, 2021; Balaji et al, 2018). The proteasome-mediated substrate destruction is mediated by an enzymatic cascade, during which ubiquitin is activated by E1, conjugated by E2 and finally covalently coupled to lysine residues of target proteins by E3 ligase. Specifically, E3 ubiquitin ligases are pivotal to the formation of ubiquitin-protein complex and subsequent target degradation by conferring substrate specificity and ubiquitin transfer (Popovic et al, 2014; Sosič et al, 2022). Recent studies have found that E3 ubiquitin ligase such as MKRN1, FBXO4, FBXL10, and RNF34 mediate protein stabilities of key metabolic regulators including AMPK, PPARγ, and PGC1α. Thus, they are tightly involved in the adipose tissue functionality and onset of obesity (Lee et al, 2018; Peng et al, 2018; Inagaki et al, 2015; Wei et al, 2012).

FBXW7 belongs to the Skp1-Cullin1- F-box protein ubiquitin ligases (SCFs) family (Lan and Sun, 2019). As a well-studied E3 ligase in cancer biology, FBXW7 was recently reported to be involved in lipid and glucose metabolism via the degradation of Fetuin-A or REV-ERBα in the liver (Zhao et al, 2018; Zhao et al, 2016). However, the role of FBXW7 in adipose tissue biology is not understood. In the present study, we found inverse correlation between FBXW7 and thermogenic fat functionality. By studying FBXW7 fat conditional transgenic and deficient mice, we demonstrated that FBXW7 promoted obesity and metabolic dysfunctions by targeting S6K1, a downstream effector of mTOR signaling, in thermogenic fat, which consequently impact preadipocyte proliferation via glycolytic product lactate. Our results suggested that adipose FBXW7 is a major regulator for thermogenic fat biology and energy homeostasis, which provides potential therapeutic target for obesity and metabolic diseases.

# Results

## FBXW7 expression is negatively correlated with thermogenic fat functionality

To evaluate the functions of FBXW7 in fat biology, we examined Fbxw7 expression levels in various metabolic organs and found that adipose tissues including BAT, inguinal WAT (iWAT), and epididymal WAT (eWAT) featured high Fbxw7 expression compared to liver and pancreas, with BAT showed the highest Fbxw7 levels (Fig. 1A; Appendix Fig. S1A). FBXW7 encodes three isoforms, including Fbxw7α, Fbxw7β, and Fbxw7γ (Matsumoto et al, 2006). We identified that Fbxw7α as the dominant isoform in adipose tissues of C57BL/6 J mice (Fig. 1B). We next evaluated whether FBXW7 may be involved in thermogenic fat functionality and obesity. Notably, FBXW7 featured significant increase in BAT and iWAT of various obese mice models, including HFD mice, leptin-deficient *ob/ob* mice, and leptin receptor-deficient *db/db* mice, and was negatively correlated with Ucp1 levels, while its expressions remained similar in eWAT (Fig. 1C; Appendix Fig. S1B–G).

In humans, subcutaneous fat shows thermogenic capacity (Bartesaghi et al, 2015; Carey et al, 2014; Overby et al, 2020). Of clinical significance, we also found that both mRNA and protein levels of FBXW7 were increased in subcutaneous WAT of obese human, and was negatively correlated with Ucp1 levels (Fig. 1D;

Appendix Fig. S1H,I), suggesting FBXW7 is involved in the suppression of thermogenic functions and the development of obesity. Moreover, examination of the relationship between FBXW7 single nucleotide polymorphisms (SNPs) and metabolic traits in human using microarray genotyping datasets from Shanghai Nicheng Cohort Study (Data ref: China National Center for Bioinformation OMIX007667, 2024) identified a genetic variant (A > G, rs1351903) in the 5'untranslated region (5'-UTR) of FBXW7 gene, with the G allele carriers characterized significantly increased body fat (Fig. 1E). The genotype distribution was 1969 subjects (23.45%) with AA, 4093 subjects (48.74%) with AG, and 2335 subjects (27.81%) with GG. The characteristics of different genotype subjects were shown in Appendix Table S1. Overall, these data suggested that FBXW7 may be involved in thermogenic functions and the development of obesity.

## Adipose-specific overexpression of FBXW7 exacerbated obesity and metabolic dysfunctions

Next, we constructed FBXW7 fat-specific transgenic mice (FBXW7-FTG mice) to mimic pathological increases of FBXW7 in adipose tissues (Appendix Fig. S2A). We observed efficient and specific overexpression of Fbxw7 with mRNA and protein levels in BAT, iWAT, and eWAT, but not in non-adipose tissues like liver, gas, and pancreas (Appendix Fig. S2B–E), indicating that the mouse model of FBXW7 fat-specific overexpression were successfully established.

FBXW7-FTG mice were similar to their wild-type (WT) littermates under chow diet (CD) in body weights, body composition, glucose tolerance, liver weights and lipid contents and serum lipid profiles (Appendix Fig. S3A–H), but showed worse insulin sensitivity (Appendix Fig. S3D). Interestingly, we observed declined whole body oxygen consumption (VO$_2$), carbon dioxide production (VCO$_2$), and energy expenditure in FBXW7-FTG mice without altered locomotor activity and food intake (Appendix Fig. S3I–K). Furthermore, we have housed mice at 30 °C for a week and subjected mice to 1 mg/kg CL-316,243 i.p. injection and measured O$_2$ consumption as described previously (Cannon and Nedergaard, 2011). Though there were no differences in oxygen consumption under thermoneutrality, FBXW7-FTG mice exhibited lower oxygen consumption upon CL-316,243 challenge, compared to their littermate controls (Appendix Fig. S3L). Besides, FBXW7-FTG mice showed reduced thermogenic capacity under cold exposure (Appendix Fig. S3M), suggesting that non-shivering thermogenesis was reduced in mice with FBXW7 overexpression in fat.

BAT is vital for energy metabolism. Detailed anatomical analysis showed that BAT of FBXW7-FTG mice were strikingly larger with a paler color compared to WT (Appendix Fig. S4A), indicative of enhanced intra-tissue lipids. Indeed, histological analysis showed that BAT characterized increased adipocyte sizes with elevated lipid deposition in adipocytes, while the numbers of nuclei per unit area in BAT were also decreased (Appendix Fig. S4B,C). Meanwhile, both mRNA and protein of UCP1 levels were decreased in BAT of FBXW7-FTG mice, suggesting impaired thermogenic capacity of BAT of FBWX7-FTG mice (Appendix Fig. S4D). Intriguingly, the genomic DNA content per depot in BAT of FBXW7-FTG mice were significantly decreased in FBXW7-FTG mice compared with WT mice (Appendix Fig. S4C), indicating the total adipocyte numbers in thermogenic fat pads of FBXW7-FTG mice may be

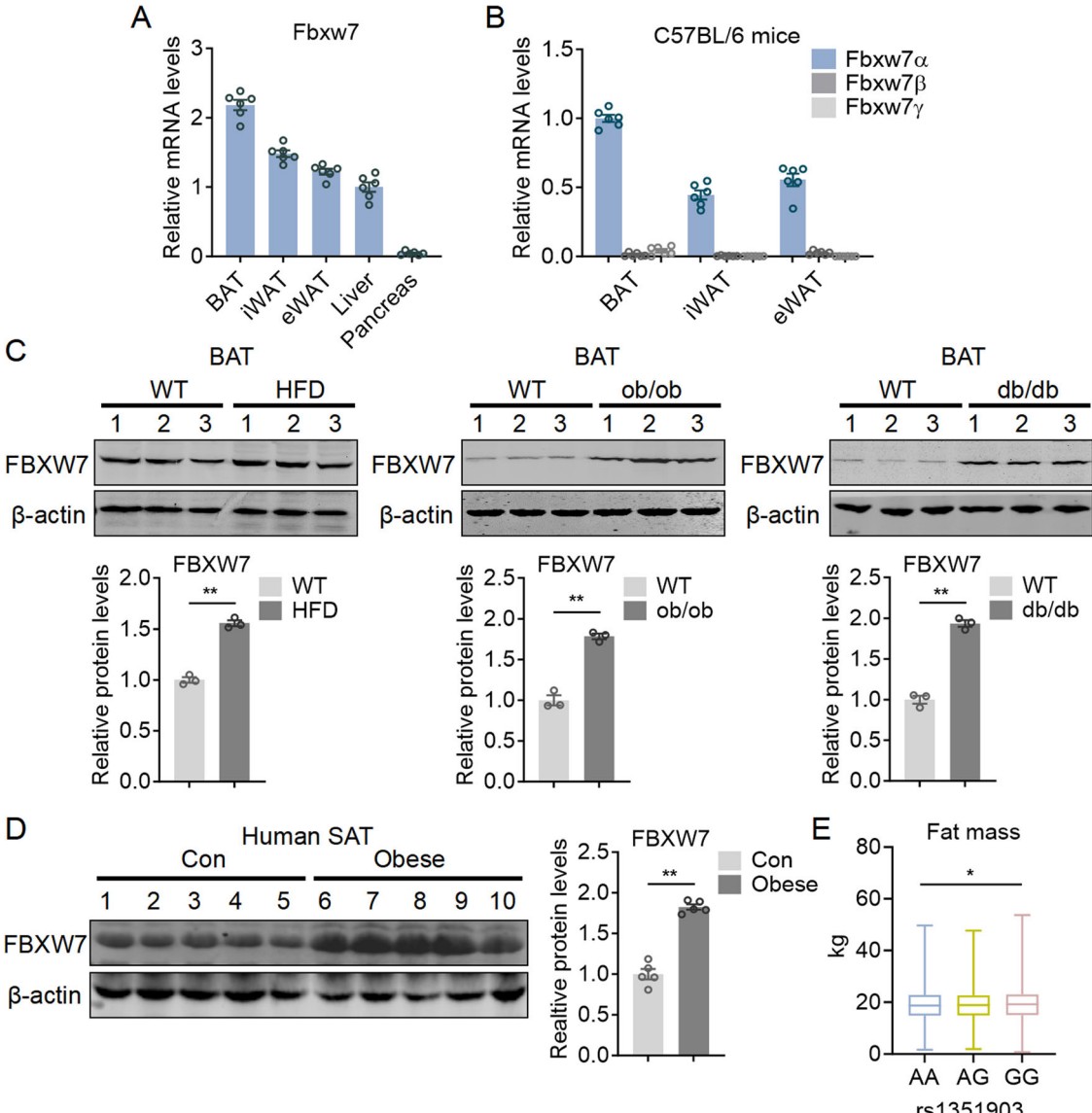

**Figure 1. Expression patterns of FBXW7 in adipose tissues and identified FBXW7 genetic variant in human.**

(A) Tissues expression profile of FBXW7. $n = 6$. (B) Relative mRNA levels of FBXW7 subtype (-α, -β, and -γ) in adipose tissues. $n = 6$. (C) Representative protein levels of FBXW7 in BAT of HFD, ob/ob and db/db mice. $n = 3$. (D) Representative protein levels of FBXW7 in SAT of control and obese subjects. $n = 5$. (E) Association between genetic variants in the FBXW7 gene region (rs1351903, AA, wild-type, $n = 1969$; AG, heterozygous, $n = 4093$; GG, mutation, $n = 2335$) with Fat mass of human. Data information: All data are representative of three individual experiments. n refers to biological replicates. (C, D) Data are presented as mean ± SEM, unpaired two-tailed Student's t-test, **$p < 0.01$. (E) Box plots: centerlines show the medians; box limits indicate the 25th and 75th percentiles; whiskers extend to the minimum and maximum. One-way ANOVA followed by Dunnett's multiple comparison test was conducted and statistical significance denoted as $p = 0.0286$. Source data are available online for this figure.

reduced. Meanwhile, compared with WT mice, the weight of iWAT was also increased in FBXW7-FTG mice, while eWAT weight was comparable (Appendix Fig. S4A), which was consistent with adipocyte sizes quantification analysis (Appendix Fig. S4E,F). Detailed analysis revealed enlarged adipocyte sizes while decreased numbers of nuclei per unit area and genomic DNA content per depot (Appendix Fig. S4G), as well as reduced both Ucp1 mRNA and protein levels in iWAT of the FBXW7-FTG mice (Appendix Fig. S4H). These data indicated that FBXW7-FTG mice characterized impaired brown fat function and reduced iWAT browning.

We then subjected these mice under energy challenge with high fat diet (HFD) feeding. After 10 weeks of HFD, HFD FBXW7-FTG mice showed increased body weights and fat mass, decreased energy expenditure and impaired thermogenic capacity after cold exposure, defective glucose and insulin tolerance, increased liver weights and lipid infiltration, as well as elevated serum lipid profiles compared to WT mice, without changes in locomotor activity and food intake (Fig. 2A–E; Appendix Fig. S5A–I). Similar to what were observed in CD (chow diet) fed mice, HFD FBXW7-FTG mice had significantly enlarged BAT depot with enlarged brown adipocyte

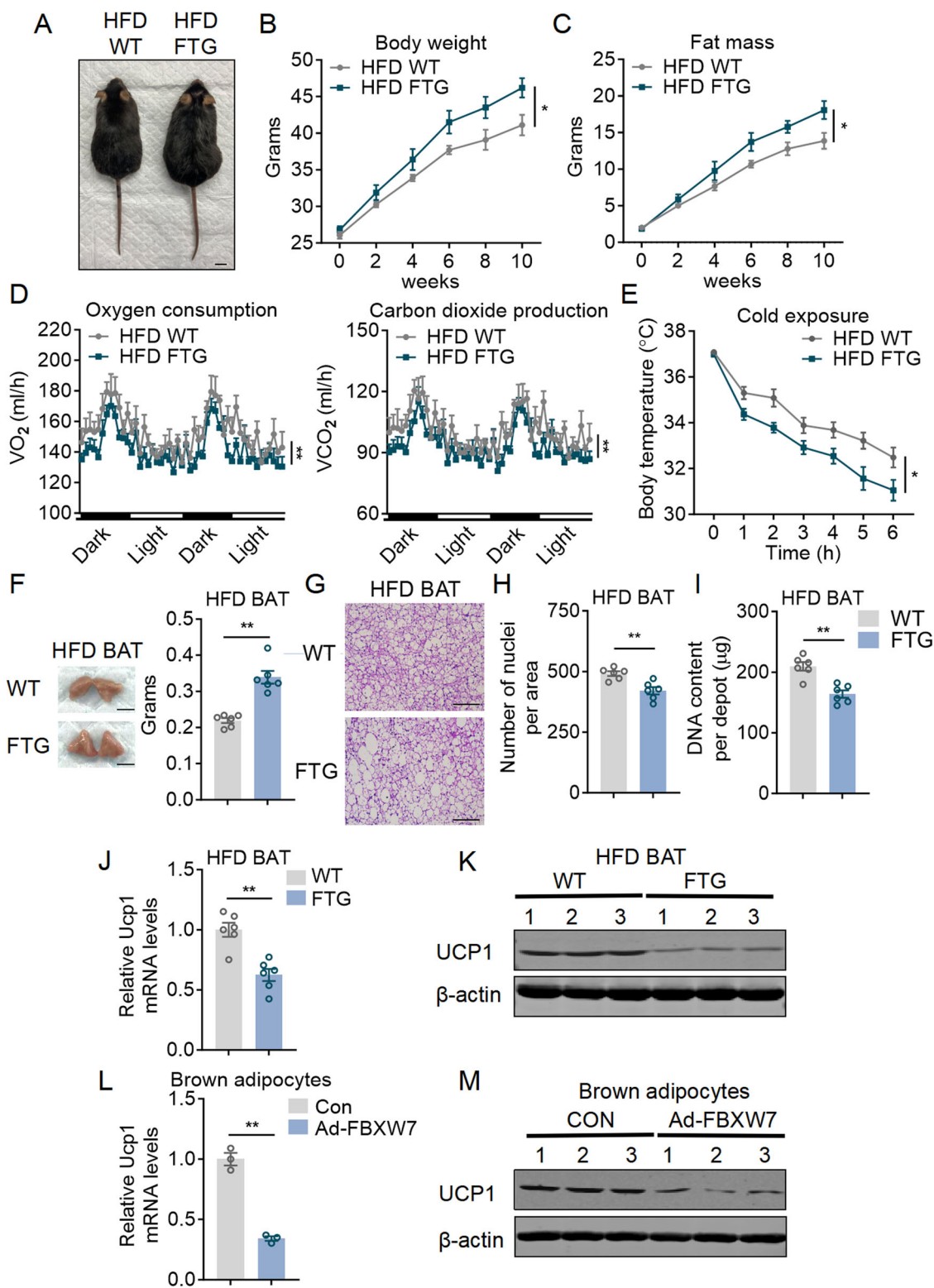

sizes, more lipid deposition and decreased numbers of nuclei per unit area (Fig. 2F–H), with decreased genomic DNA content per depot (Fig. 2I). Moreover, Both Ucp1 mRNA and protein expressions were reduced in BAT of FBXW7-FTG mice (Fig. 2J,K). These effects were largely caused by the high FBXW7 levels in brown adipocytes, as FBXW7 overexpression via adenoviral delivery in brown adipocytes also decreased UCP1 levels (Fig. 2L,M). The iWAT and eWAT of HFD FBXW7-FTG mice were also increased with enhanced adipocyte sizes, suggesting more lipid accumulation caused by impaired energy expenditure

**Figure 2.  Fat FBXW7 overexpressed mice impaired brown fat functionality and decreased energy expenditure under HFD.**

(**A–M**) Metabolic performances of high fat diet fed WT and FBXW7-FTG mice. (**A–C**) Morphological appearance, body weight and fat mass; scale bar represents 1 cm, $n = 6$. (**D**) Energy expenditure as shown by oxygen consumption and carbon dioxide production of mice at an ambient temperature of 22 °C, $n = 5$. (**E**) Rectal temperatures of mice during 6 h cold exposure at 4 °C, $n = 6$. (**F**) Weight of brown fat pads (BAT), scale bar represents 1 cm, $n = 6$. (**G**) Representative images of H&E staining of BAT, scale bar represents 100 μm. (**H, I**) Nuclei densities and genomic DNA content in BAT, $n = 6$. (**J–M**) Relative mRNA and protein level of Ucp1 in BAT of WT and FBXW7-FTG mice, $n = 6$ and (**L, M**) in brown adipocytes infected with ADV-GFP or ADV-FBXW7 for 2 days, $n = 3$. Data information: All data are representative of three individual experiments. $n$ refers to biological replicates. (**B–E**) Data are presented as mean ± SEM. Two-way ANOVA followed by Sidak's multiple comparison test was conducted and statistical significance denoted as *$p < 0.05$ (**B**: $p = 0.0408$; **C**: $p = 0.0458$; **D**: Left panel: $p < 0.0001$; **D**: Right panel: $p < 0.0001$; **E**: $p = 0.0198$). (**F, H–J, L**), Data are presented as mean ± SEM, unpaired two-tailed Student's t-test, **$p < 0.01$. Source data are available online for this figure.

(Appendix Fig. S5J–L). Interestingly, we also observed decreased numbers of nuclei per unit area and genomic DNA content per depot, as well as reduced UCP1 expressions in iWAT of FBXW7-FTG mice (Appendix Fig. S5M). In addition, FBXW7 over-expression in beige fat or beige adipocytes also suppressed UCP1 levels, suggesting possible cell-autonomous effects of FBXW7 in the regulation of beige fat function (Appendix Fig. S5N,O). Taken together, these data suggest that FBXW7 aggravates obesity and metabolic dysfunctions in mice, possibly due to impaired tissue expansion and thermogenesis in thermogenic fat.

## Adipose-specific ablation of FBXW7 induces BAT expansion and browning of white fat, thus protects against obesity in mice

Conversely, we next generated FBXW7 fat-conditional knockout mice (FBXW7-FKO mice) by crossing FBXW7-LoxP mice with Adiponectin-Cre mice (Appendix Fig. S6A). We confirmed efficient and specific deletion of Fbxw7 with mRNA and protein levls in BAT, iWAT, and eWAT, but not in non-adipose tissues including liver, gastrocnemius muscle or pancreas, of FBXW7-FKO mice (Appendix Fig. S6B–E), indicating that the mouse model of FBXW7 fat-specific knockout were successfully established.

FBXW7-FKO mice fed under CD were similar to their WT littermates with comparable body weights, body composition, insulin sensitivity, liver weights and lipid contents and serum parameters (Appendix Fig. S7A–H). We observed increased energy expenditure in FBXW7-FKO mice compared to WT mice, without changes in locomotor activity and food intake (Appendix Fig. S7I–K). Furthermore, FBXW7-FKO mice exhibited high oxygen consumption upon CL-316,243 administration, compared to their separate littermate controls (Appendix Fig. S7L). FBXW7-FKO mice also showed enhanced thermogenic capacity under cold exposure (Appendix Fig. S7M), suggesting that non-shivering thermogenesis was enhanced in mice with FBXW7 deletion in fat.

Detailed analysis showed that in contrast with BAT phenotype of FBXW7-FTG mice, FBXW7-FKO mice had significantly larger BAT compared to WT (Appendix Fig. S8A). This was not due to increased lipid deposition in brown adipocytes as in FBXW7-FTG mice, as BAT of FBXW7-FKO mice featured similar lipid contents and number of nuclei per area in histological analysis, indicating their brown adipocytes were of similar cellular sizes (Appendix Fig. S8B). Instead, contrary to FBXW7-FTG mice, we found that DNA content per depot of FKO BAT was significantly increased, suggesting the sharply increased BAT weight were mainly due to increased cell numbers caused by enhanced BAT expansion (Appendix Fig. S8C). Consistent with their enhanced energy expenditure, BAT and primary brown adipocytes of FBXW7-FKO

mice both showed increased both Ucp1 mRNA and protein expression per unit and per depot compared to WT (Appendix Fig. S8D,E). In addition, hematoxylin-Eosin (H&E) staining and cross-sectional area (CSA) quantification analysis showed significantly reduced lipid droplets, while increased the numbers of nuclei per unit area and genomic DNA content per depot, as well as increased both Ucp1 mRNA and protein levels in iWAT of FBXW7-FKO mice compared to those in WT mice (Appendix Fig. S8F–I), while there was no differences in adipocyte sizes in eWAT (Appendix Fig. S8F,G). These results revealed that FBXW7-FKO mice exhibit BAT expansion and iWAT browning for enhanced energy expenditure and thermogenesis.

Considering that brown fat specific FBXW7-BKO mice may provide more direct evidences to demonstrate the role of FBXW7 in the regulation of BAT physiology and thermogenesis and energy homeostasis, We have utilized an adeno-associated virus carrying adipoQ promoter driven shFBXW7 or control to inject locally in BAT to examine the phenotypes of FBXW7 specific deficiency in BAT. The results showed that, similarly with FBXW7-FKO mice, FBXW7 knockdown in BAT resulted in increased energy expenditure upon CL-316,243 treatment and enhanced thermogenic capacity after cold exposure with similar body weights and body composition under chow diet (Appendix Fig. S9A–D). Detailed analysis showed that FBXW7 knockdown in BAT of mice showed significantly larger BAT compared to control mice (Appendix Fig. S9E). Importantly, we found that DNA content per depot was significantly increased in BAT with FBXW7 knockdown, while lipid contents and numbers of nuclei per area were similar in histological analysis (Appendix Fig. S9F–H), suggesting the increased BAT weights were mainly due to increased cell numbers (Appendix Fig. S9H). Consistent with enhanced energy expenditure, FBXW7 knockdown in BAT showed increased Ucp1 expression compared to control group (Appendix Fig. S9I,J). These results revealed that FBXW7 knockdown in BAT of mice cuased BAT expansion for enhanced energy expenditure and thermogenesis, similarly as those in FBXW7-FKO mice.

We then subjected FBXW7-FKO mice under HFD feeding. HFD FKO mice exhibited reduced body weights and fat mass compared to HFD WT (Fig. 3A–C; Appendix Fig. S10A). Energy expenditure analysis showed that oxygen consumption ($VO_2$), carbon dioxide production ($VCO_2$) and heat were significantly increased in HFD FBXW7-FKO mice, with consistently better maintenance of core temperature in acute cold challenge (Fig. 3D,E; Appendix Fig. S10B), without alternations in locomotor activity and food intake (Appendix Fig. S10C). HFD FBXW7-FKO mice had ameliorated hepatic steatosis and serum lipid profiles compared to HFD WT, while their glucose and insulin sensitivity remained largely the same (Appendix Fig. S10D–I). Interestingly, FBXW7-

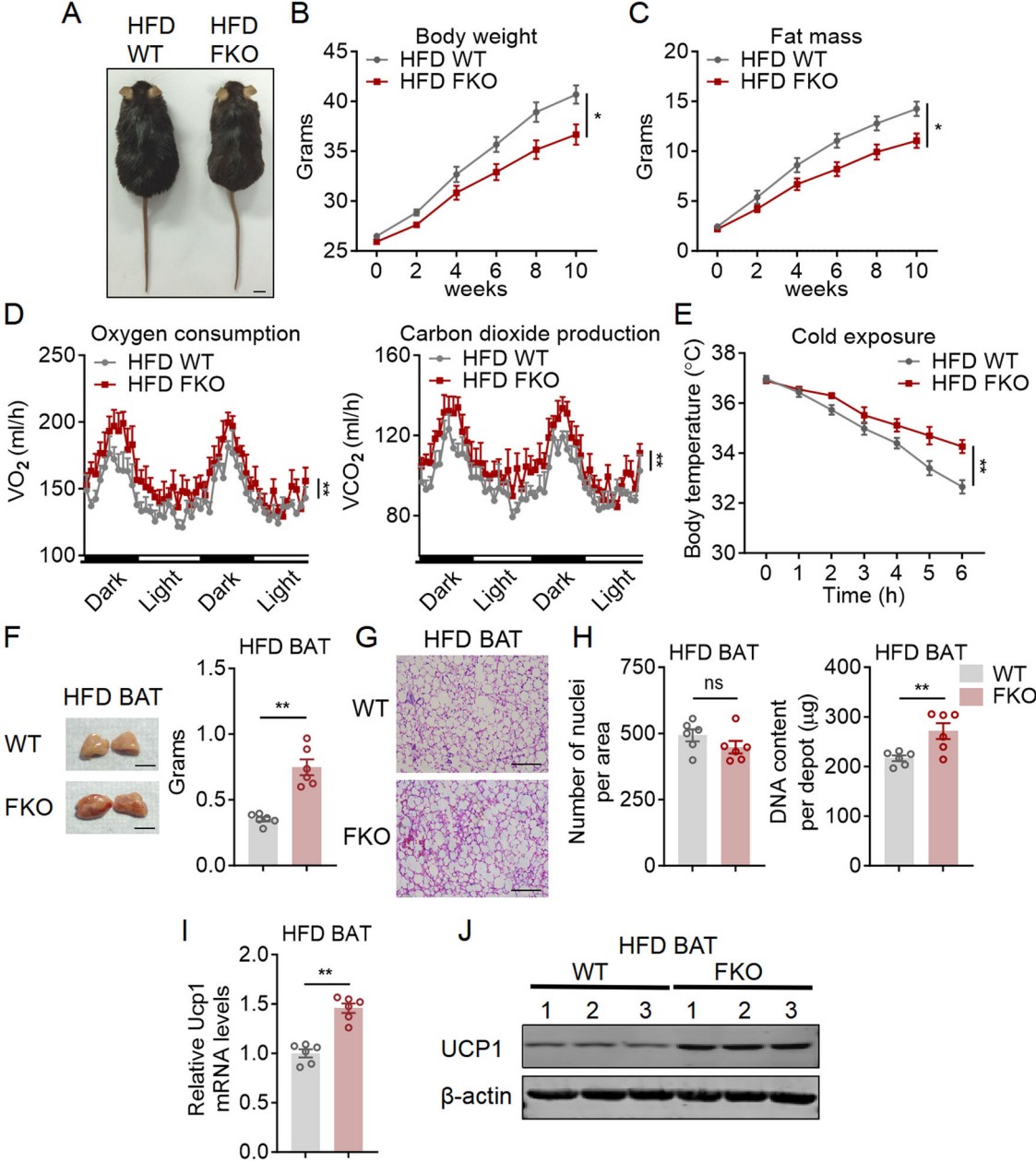

**Figure 3. Fat FBXW7 deficient mice led to brown fat expansion and increased energy expenditure under HFD.**

(A–J) Metabolic performances of high fat diet fed WT and FBXW7-FKO mice. (A–C) Morphological appearance, Body weight and fat mass, scale bar represents 1 cm, $n = 6$; (D) Energy expenditure as shown by oxygen consumption and carbon dioxide production of mice at an ambient temperature of 22 °C, $n = 5$. (E) Rectal temperatures of mice during 6 h cold exposure at 4 °C, $n = 6$. (F) Weight of brown fat pads (BAT), scale bar represents 1 cm, $n = 6$. (G) Representative images of H&E staining of BAT, scale bar represents 100 μm. (H) Nuclei densities and genomic DNA content in BAT, $n = 6$. (I, J) Relative mRNA and protein level of Ucp1 in BAT of WT and FBXW7-FKO mice, $n = 6$. Data information: All data are representative of three individual experiments. $n$ refers to biological replicates. (B–E) Data are presented as mean ± SEM. Two-way ANOVA followed by Sidak's multiple comparison test was conducted and statistical significance denoted as $*p < 0.05$ and $**p < 0.01$ (B: $p = 0.0144$; C: $p = 0.0192$; D: Left panel: $p < 0.0001$; D: Right panel: $p < 0.0001$; E: $p = 0.0083$). (F–I) Data are presented as mean ± SEM, unpaired two-tailed Student's t-test, $**p < 0.01$, ns: not significant. Source data are available online for this figure.

FKO mice showed impaired insulin signaling as shown by decreased p-PI3K, p-AKT, and p-GSK3β levels in BAT and iWAT. However, the insulin signaling of the liver and muscle was improved as shown by increased p-PI3K, p-AKT, and p-GSK3β levels, possibly due to reduced body weights (Appendix Fig. S10J). The combined effects of insulin sensitivity in these tissues may

explain the unaltered whole-body glucose metabolism in WT and FBXW7 FKO mice.

In addition, detailed analysis showed that, similar to FKO mice under CD, the gross appearance of HFD FKO BAT was much larger with an apparent brownish color (Fig. 3F), indicative of heightened thermogenic function. Histological analysis showed that although

HFD FKO BAT were significantly larger, its BAT characterized similar lipid infiltration and numbers of nuclei per area (Fig. 3G,H), while genomic DNA content per depot were significantly increased (Fig. 3H), indicating the larger BAT in HFD FBWX7-FKO mice were due to enhanced BAT expansion, while the reddish color was possibly caused by elevated thermogenesis, i.e., UCP1 (Fig. 3I,J). Moreover, iWAT and eWAT of HFD FBXW7-FKO mice were significantly smaller with reduced adipocyte sizes, with decreased numbers of nuclei per unit area and genomic DNA content per depot, as well as increased both Ucp1 mRNA and protein expressions in iWAT of FBXW7-FKO mice, possibly caused by both cell-autonomous effects in beige fat and elevated BAT functionality and energy expenditure of these mice (Appendix Fig. S10K–O). Taken together, these data suggested that FBXW7 deficiency in adipose tissues protected mice from HFD induced obesity, possibly due to the combined effects of enhanced BAT expansion and browning of white fat for enhanced thermogenesis and energy expenditure.

## FBXW7 deficiency in fat enhanced ribosomal protein synthesis and S6K1 protein level

We then set out to elucidate the mechanism of FBXW7 in the regulation of fat biology and energy homeostasis. Since BAT manifested a very strong phenotype, we used BAT from WT and FBXW7-FKO mice under HFD and subjected them to mass spectrometry analysis (Dataset EV1). KEGG pathway analysis on differentially expressed proteins revealed ribosomal proteins were significantly increased in BAT of FBXW7-FKO mice (Fig. 4A). Consistently, GO enrichment analysis also manifested that top changes were translation and ribosome-related events (Appendix Fig. S11A). mTOR signaling has been well documented to promote ribosome protein synthesis (Saxton and Sabatini 2017). It has also been reported that mTOR is actively involved in regulation of tissue expansion as well as energy homeostasis, including browning of white fat (Mannaa et al, 2013; Albert and Hall, 2015). In addition, we performed immunoprecipitation coupled with mass spectrum (IP-MS) to identify the interacting proteins of FBXW7 in FBXW7 overexpressed brown adipocytes (Dataset EV2). Of note, Venn diagram analysis of LC-MS/MS and IP-MS results suggested that S6K1, a key protein in mTOR signaling, is a potential FBXW7 substrate (Fig. 4B). Further comparison of various key mTOR signaling proteins, including mTOR, Rictor, Raptor, and S6K1, in the proteomics also pointed out that S6K1 was uniquely upregulated in BAT of FBXW7-FKO mice (Fig. 4C). Indeed, S6K1 and phosphorylated S6K1 (p-S6K1) protein levels were increased in BAT of FBXW7-FKO mice while decreased in BAT of FBXW7-FTG mice, with phosphorylated RPS6 (p-S6) protein levels as positive control (Fig. 4D,E, left panel, Appendix Fig. S11B), but their S6K1 mRNA levels remained similar compared to WT (Fig. 4D,E, right panel), indicating that the changes in S6K1 protein levels were majorly on translational levels. Similar trends of changes in S6K1 protein levels were observed in primary brown adipocytes with FBXW7 knockout or overexpression, without changes in mRNA levels (Fig. 4F–H). Meanwhile, FBXW7 knockout or overexpression in iWAT and beige adipocytes also increased or suppressed S6K1 and phosphorylated S6K1 (p-S6K1) protein levels, without altered S6K1 mRNA levels, suggesting the FBXW7-S6K1 regulatory axis also exist in iWAT (Appendix Fig. S11C–F).

In addition, it has been reported that S6K1 phosphorylated IRS1 at S307 site to inhibit the binding of IRS1 to PI3K, which leads to impaired insulin signaling (Um et al, 2004). Our results showed that p-IRS1 (S307) levels were reduced in BAT and iWAT of FBXW7-FTG mice, while increased in BAT and iWAT of FBXW7-FKO mice, suggesting that S6K1 downstream targets were changed accordingly (Appendix Fig. S11G).

Overall, these data suggested that FBXW7 might modulate mTOR signaling pathway via its regulation on S6K1 protein levels.

## FBXW7 negatively targets S6K1 for ubiquitination and degradation

FBXW7 mediates covalent binding of ubiquitin on protein substrates and thus targets it for proteosome degradation (Lan and Sun, 2019). We thus examined whether S6K1 is a FBXW7 substrate and whether FBXW7 targets S6K1 for ubiquitination and proteasome degradation. Indeed, recombinant FBXW7 protein-mediated S6K1 protein ubiquitination in vitro (Fig. 5A) and co-immunoprecipitation assays showed that FBXW7 physically interacted with S6K1 in overexpressed HEK293T and endogenously in BAT and brown adipocytes (Fig. 5B,C; Appendix Fig. S11H). Critically, we identified the presence of a consensus FBXW7 substrate motif in S6K1 (Yumimoto and Nakayama, 2020), which is conserved in various species (Appendix Fig. S11I). FBXW7 functions together with Skp1 and Cullin1 (SCF complex) to form functional E3 ligase and exert substrate ubiquitination and subsequent degradation (Chen et al, 2015). Treatment of cells with MLN4924, a pan Cullin protein family inhibitor, inhibited NEDD8 as previously reported (Blank et al, 2013) and blocked S6K1 degradation in a dose-dependent manner, suggesting the involvement of the SCF complex in S6K1 degradation (Fig. 5D). Dominant-negative Cullin1 expression specifically blunted S6K1 degradation, further supporting the involvement of the SCF complex (Fig. 5E). In addition, an in vivo ubiquitination assay in overexpressed HEK293T and endogenously brown adipocytes demonstrated that FBXW7 mediates S6K1 ubiquitination. MG132 treatment increased ubiquitination, which served as a positive control (Fig. 5F; Appendix Fig. S11J). To investigate the effect of FBXW7 on S6K1 protein stability, we treated cells with cyclohex-imide (CHX) to inhibit new protein synthesis. FBXW7 over-expression significantly promoted S6K1 degradation in a time-dependent manner (Fig. 5G). Importantly, co-expression of FBXW7 and S6K1 enhanced S6K1 degradation, which was blocked by the proteasome inhibitor MG132 (Fig. 5H). Taken together, our findings indicate that FBXW7 mediates the covalent binding of ubiquitin on S6K1 protein substrates and targets it for proteasome degradation through the SCF complex.

## S6K1 mediates mature adipocytes FBXW7 deficiency induced lactate production and hyperplasia of brown preadipocytes

Since we found that FBXW7 deficiency increased brown fat sizes and functions by improving adipocyte expansion and thermogenesis, to further understand the intrinsic mechanisms, we performed flow cytometry analysis using SVF derived from BAT depot of FBXW7-FKO and WT mice. We found increased percentage of APCs (CD45⁻CD31⁻Sca1⁺ cells) from FBXW7-FKO mice (Fig. 6A).

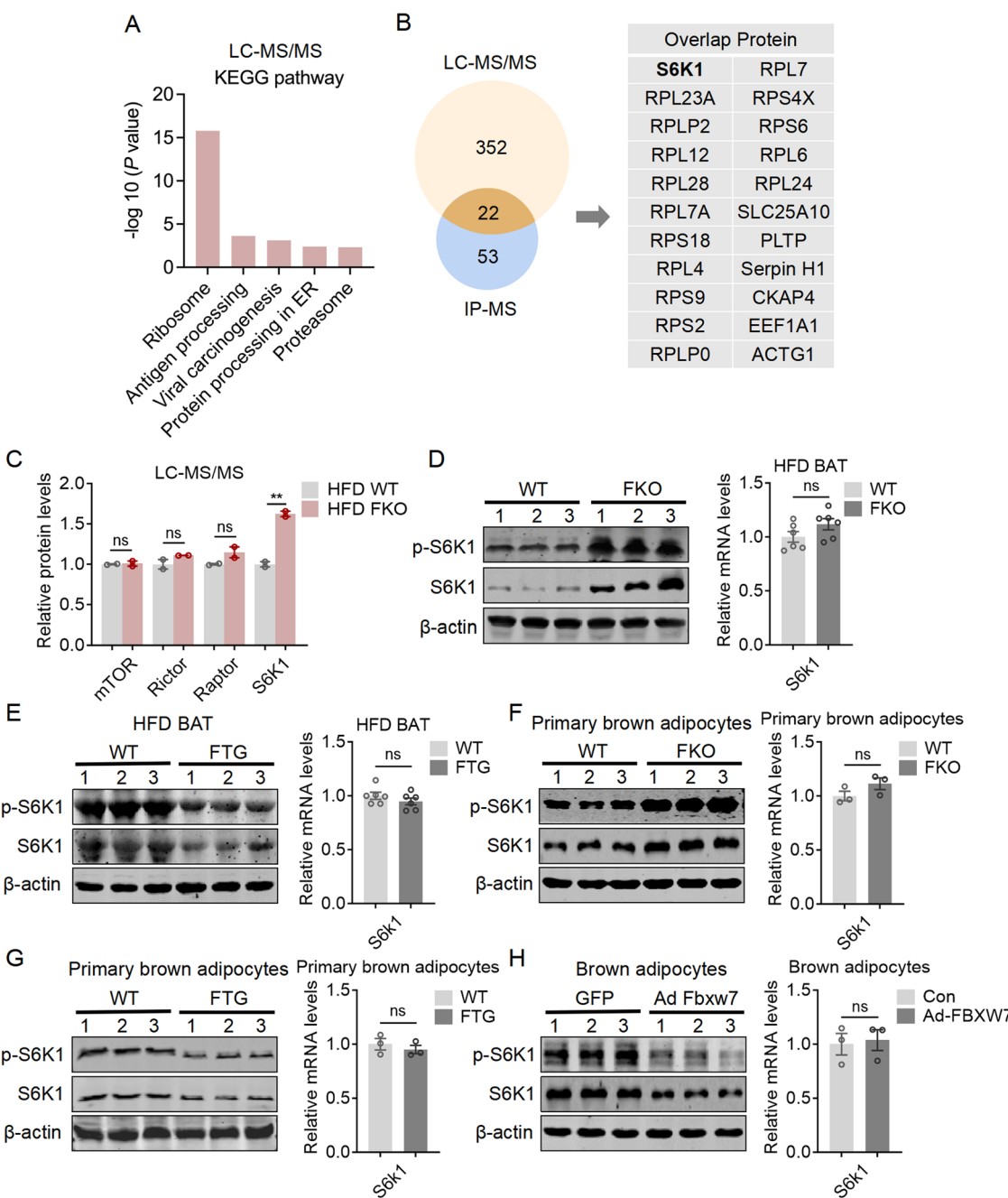

**Figure 4. FBXW7 deficiency in fat enhanced ribosomal protein synthesis and S6K1 protein level.**

(A, B) KEGG analysis of BAT protein in LC-MS/MS analysis of WT and FBXW7-FKO mice fed with high fat diet. (B) Venn diagram analysis of LC-MS/MS of WT and FBXW7-FKO mice and IP-MS of FBXW7 interacted proteins in FBXW7 overexpressed brown adipocytes. (C) Relative protein levels of mTOR signaling pathway in LC-MS/MS analysis, $n = 2$. (D–H) p-S6K1 and S6K1 protein levels, as well as relative mRNA levels of S6K1 in BAT from WT, FBXW7-FKO, or FBXW7-FTG mice fed with high fat (D, E), $n = 6$; in primary brown adipocytes from WT, FBXW7-FKO or FBXW7-FTG mice (F, G), $n = 3$; in brown adipocytes infected with ADV-GFP or ADV-FBXW7 for 2 days (H), $n = 3$. Data information: All data are representative of three individual experiments. $n$ refers to biological replicates. (A) KEGG enrichment analysis was performed with the R package clusterProfiler, with a Bonferroni correction and an adjusted $p$-value of 0.05. (C–H) Data are presented as mean ± SEM, unpaired two-tailed Student's t-test, **$p < 0.01$. ns: not significant. Source data are available online for this figure.

Moreover, FKO APCs had upregulated Ki67 mRNA levels compared to WT APCs (Fig. 6B), suggesting FBXW7-FKO mice had more brown preadipocytes due to enhanced proliferation, which may underlie the significantly larger BAT depots and increased DNA content per unit we observed in FBXW7-FKO mice.

mTOR signaling is vital for glycolysis (Fan et al, 2021; Ramirez-Moral et al, 2021; Gu et al, 2015). Lactate, the key product of glycolysis, has been shown to increase brown preadipocytes proliferation (Wang et al, 2020). We found significantly increased glycolytic gene programs in BAT and primary brown adipocytes of

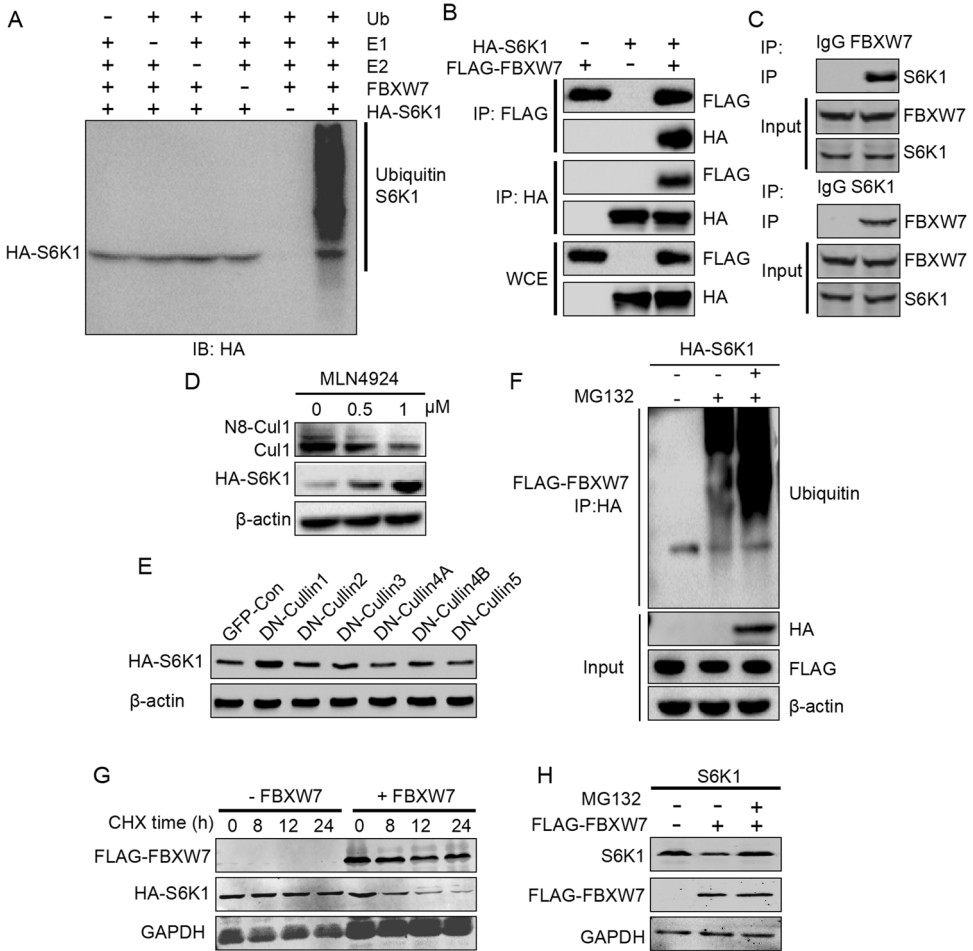

**Figure 5.** **FBXW7 directly interacts with S6K1 and mediates ubiquitination of S6K1.**

(A) In vitro ubiquitination of HA-S6K1 in the presence or absence of recombinant His-FBXW7 protein, E1, E2, ubiquitin, and an ATP-regeneration system. (B) Co-immunoprecipitation of Flag-FBXW7 and HA-S6K1 in 293T cells. (C) Co-Immunoprecipitation of FBXW7 and S6K1 in brown fat tissue of mice. (D) The time course of HA-S6K1 and NEDD8 levels in 293T cells stably expressing HA-S6K1 treated with MLN4924, a pan Cullin protein family inhibitor. (E) Western blot analysis of HA-S6K1 levels in 293T cells transfected with different dominant-negative (DN) Cullin family proteins. (F) Ubiquitination assay of HA-S6K1 in 293T cells with or without Flag-FBXW7 overexpression and the proteasome inhibitor MG132. (G) The time course of HA-S6K1 levels in 293T cells treated with cycloheximide (CHX) to inhibit protein synthesis, with or without Flag-FBXW7 overexpression. (H) Degradation of endogenous S6K1 by Flag-FBXW7 with or without the proteasome inhibitor MG132. Source data are available online for this figure.

HFD FBXW7-FKO mice compared to WT, as well increased lactate levels in BAT of HFD FBXW7-FKO mice (Fig. 6C,D; Appendix Fig. S12A), Moreover, the glycolytic genes and lactate levels were inhibited in BAT from FBXW7-FTG mice (Appendix Fig. S12B,C).

Lactate promoted brown preadipocytes proliferation in a dose-dependent manner (Appendix Fig. S12D–F). Importantly, pre-adipocytes proliferation were enhanced when treated with culture medium from mature brown adipocytes of FBXW7-FKO mice compared to WT mice, which were blocked by S6K1 inhibitor PF4708671 or S6K1 siRNA treatment, as shown by EdU staining, CCK8 assay and Ki67 levels (Fig. 6E–I). In addition, these effects were also blocked by lactate transporter inhibitor AZD3965, as shown by EdU staining, CCK8 assay, and Ki67 levels (Appendix Fig. S12G–I). Meanwhile, lactate levels changed accordingly in culture medium from mature adipocytes (Appendix Fig. S12J), suggesting that FBXW7 deficiency in brown adipocytes led to

increased brown preadipocyte proliferation via enhanced lactate secretion. In contrast to that in FBXW7-FKO mice, preadipocyte proliferation was decreased when treated with culture medium from mature brown adipocytes of FBXW7-FTG mice compared to WT mice, as shown by EdU staining, CCK8 assay, and Ki67 levels (Appendix Fig. S12K–M), accompanied with reduced lactate levels (Appendix Fig. S12N).

It has been reported that mTOR signaling and S6K1 are both shown to regulate the transcription factor hypoxia-inducible factor 1 alpha (HIF-1α) for glycolytic reprogramming, including Pkm2 transcription (Sun et al, 2011; Tandon et al, 2011; Lee et al, 2019). Indeed, we showed that S6K1 siRNA or S6K1 specific inhibitor PF4708671 treatment blocked FBXW7 deficiency-induced glycolysis (Appendix Fig. S12O). Moreover, preadipocytes proliferation was increased when treated with culture medium from mature brown adipocytes of FBXW7-FKO mice compared to WT mice,

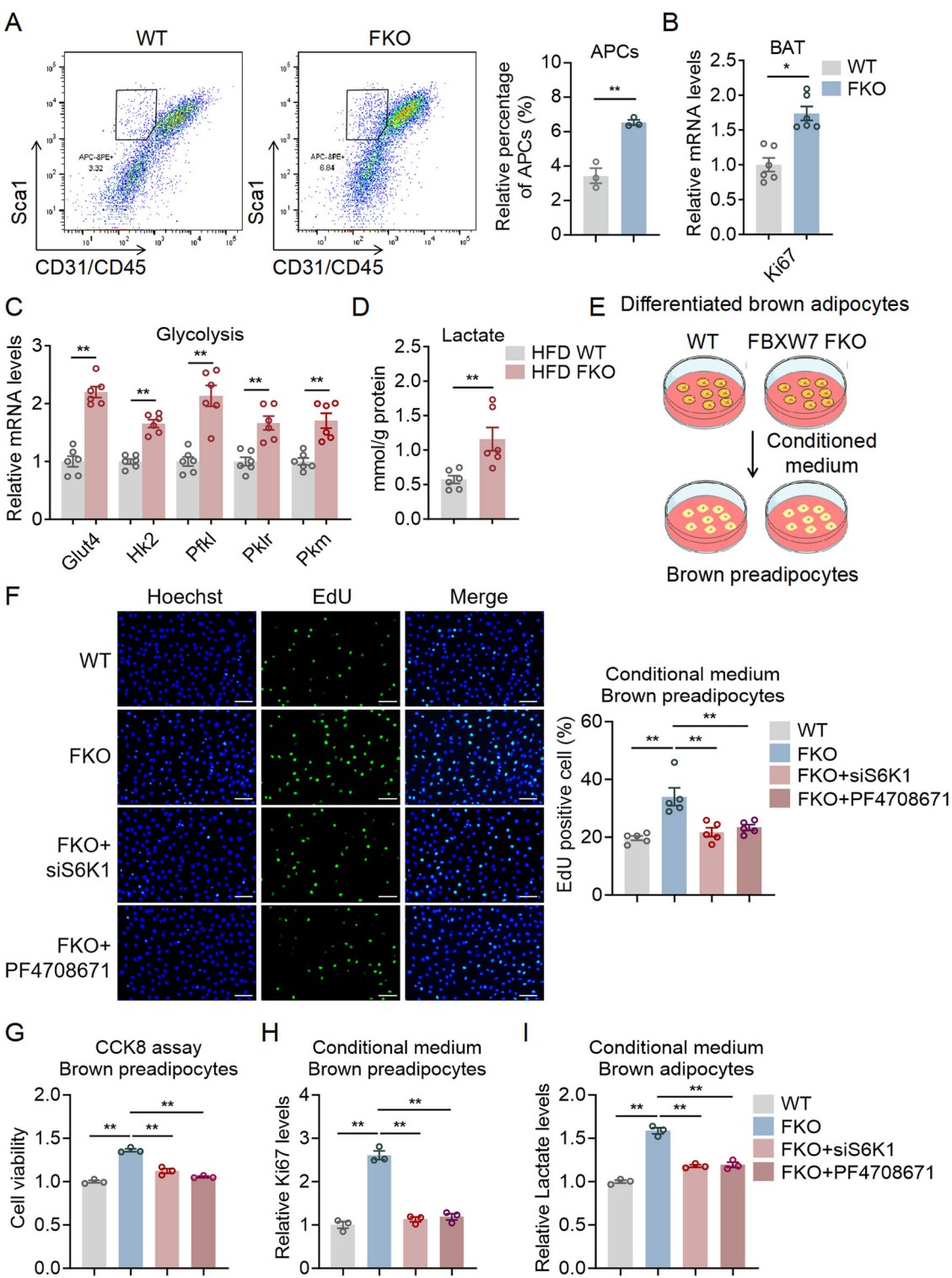

which effects were blocked by HIF-1α inhibitor PX-478, as shown by EdU staining, CCK8 assay, and Ki67 levels (Appendix Fig. S12P–S). In addition, lactate levels were changed accordingly, suggesting that FBXW7 KO leads to enhanced glycolytic reprogramming and increased brown preadipocyte proliferation through S6K1/HIF1α axis.

## S6K1 acts downstream of FBXW7 for the regulation of thermogenic fat function and obesity

In addition, to further strengthen the functional relevance between FBXW7 and S6K1 in brown and beige fat function and energy metabolism in vivo, we performed rescue experiments by

**Figure 6. FBXW7 deficiency in fat promotes lactate production, which mediates brown preadipocyte proliferation.**

(A) Representative cell sorting and quantification of the percentage of Sca-1[+] cells from BAT SVF of WT and FBXW7-FKO mice by FACS (CD45[-]/CD31[-]), $n = 3$. (B) Relative mRNA level of Ki67 in BAT from WT and FBXW7-FKO mice, $n = 6$. (C, D) Relative mRNA level of glycolytic genes and lactate levels in BAT from WT and FBXW7-FKO mice fed with high fat diet, $n = 6$. (E) Representative images of co-culture experimental procedure. (F) EdU staining, scale bar represents 50 μm, $n = 5$; (G) cell viability, $n = 3$; and (H) mRNA level of Ki67 of brown preadipocytes treated with the culture medium of differentiated brown adipocytes from SVF in WT and FBXW7-FKO mice pretreated with S6K1 siRNA or S6K1 inhibitor-PF4708671 at 10 μM, $n = 3$. (I) Relative lactate level in the culture medium of differentiated brown adipocytes from SVF in BAT of WT and FBXW7-FKO mice pretreated with S6K1 siRNA or S6K1 inhibitor-PF4708671 at 10 μM, $n = 3$. Data information: All data are representative of three individual experiments. $n$ refers to biological replicates. (A–D) Data are presented as mean ± SEM, unpaired two-tailed Student's t-test, *$p < 0.05$, **$p < 0.01$. (F–I) Data are presented as mean ± SEM. One-way ANOVA followed by Dunnett's multiple comparison test was conducted and statistical significance denoted as **$p < 0.01$ (F: $p = 0.0002$; G: $p < 0.0001$; H: $p < 0.0001$; I: $p < 0.0001$). Source data are available online for this figure.

administrating AAV carrying adipoQ promoter driven shS6K1 construct (shAAV-S6K1) and control in BAT and iWAT of WT and FBXW7-FKO mice. The shAAV-S6K1 AAV delivery achieved specific S6K1 reduction in BAT and iWAT in FBXW7-FKO mice and comparable levels in WT mice (Appendix Fig. S13A). When subjected to HFD feeding, FBXW7-FKO mice with fat-S6K1 knockdown showed increased body weights and fat mass, decreased energy expenditure and impaired thermogenic capacity after cold exposure, increased liver weights and lipid infiltration, as well as elevated serum lipid profiles compared to FBXW7-FKO mice, without changes in locomotor activity and food intake (Fig. 7A–E; Appendix Fig. S13B–H). In addition, HFD fed FBXW7-FKO mice with fat-S6K1 knockdown showed significantly enlarged BAT depot with increased brown adipocyte sizes, more lipid deposition, while decreased numbers of nuclei per unit area (Fig. 7F,G), genomic DNA content per depot (Fig. 7H) and Ucp1 mRNA levels (Fig. 7I), compared to those of FBXW7-FKO mice.

Meanwhile, the weights of iWAT and eWAT of HFD FBXW7-FKO mice with fat-S6K1 knockdown were enhanced with increased adipocyte sizes (Appendix Fig. S13I–K), decreased numbers of nuclei per unit area and genomic DNA content per depot, as well as reduced Ucp1 mRNA levels in iWAT, compared to those of FBXW7-FKO mice (Appendix Fig. S13L,M). It has been shown that mTOR signaling increased Pgc1a and Ucp1 gene transcription both in adipocytes and muscles (Xu et al, 2021; Winther et al, 2018). Besides, glycolysis likely supports thermogenesis by providing pyruvate and other intermediates to fuel the TCA cycle and oxidative phosphorylation, which may increase Ucp1. Thus, in order to understand whether S6K1 is also responsible Ucp1 levels in mature brown adipocytes, we treated mature brown adipocytes from WT and FBXW-FKO mice with S6K1 inhibitor PF4708671 and found that S6K1 inhibition reduced Ucp1 levels and blocked FBXW7-deficiency induced Ucp1 induction in brown adipocytes (Appendix Fig. S13N).

Furthermore, we have added S6K1 knockdown group with injection of AAV-CON or AAV-shS6K1 in both BAT and iWAT of control mice. The results showed that local administration of AAV-shS6K1 increased body weights and fat mass, decreased energy expenditure and impaired thermogenic capacity after cold exposure without changes in locomotor activity and food intake (Appendix Fig. S14A–H). Detailed analysis showed that, similar to FTG mice under HFD, HFD fed WT mice with fat-S6K1 knockdown showed significantly enlarged BAT depot with increased brown adipocyte sizes, more lipid deposition, while decreased numbers of nuclei per unit area (Appendix Fig. S14I–K), genomic DNA content per depot (Appendix Fig. S13K) and Ucp1 mRNA and protein levels (Appendix Fig. S14L), compared to those of WT mice. Meanwhile,

the weights of iWAT and eWAT of HFD mice with fat-S6K1 knockdown were enhanced with increased adipocyte sizes (Appendix Fig. S14M–O), decreased numbers of nuclei per unit area and genomic DNA content per depot, as well as reduced Ucp1 mRNA and protein levels in iWAT, compared to those of HFD WT mice (Appendix Fig. S14P,Q). In addition, HFD fat-S6K1 knockdown mice showed increased liver weights and lipid infiltration, as well as elevated serum lipid profiles compared to WT mice (Appendix Fig. S14R–U). Taken together, these data suggested that both brown fat expansion and Ucp1 levels were regulated by S6K1.

Overall, these data suggested that both brown fat expansion and Ucp1 levels were regulated by FBXW7-S6K1 axis, which may both contributed to the effects of FBXW7 upregulation on energy expenditure and thermogenesis.

## Discussion

Thermogenic fat play critical roles in energy homeostasis to defend progression of obesity and metabolic diseases. Although transcriptional regulation of thermogenic fat functionality has been extensively studied, the contribution of post-translational modification, particularly ubiquitination modification for modulating protein levels, is not well understood. In this study, we investigate the function of E3 ubiquitin ligase FBXW7 in fat biology. Considering the high expression of FBXW7 in brown fat which exerts critical influences on whole-body energy homeostasis, we majorly focused on the role of FBXW7 in brown fat biology. Indeed, FBXW7 affects DNA contents of brown fat and its deficiency promotes a striking two-folds increases in brown fat weights both under chow diet and high fat diet. Mechanistically via proteomics and molecular studies, we found that FBXW7 affected ribosome synthesis and glycolysis by specifically targeting S6K1, a downstream effector of mTOR signaling in mature adipocytes, which then induced preadipocytes proliferation via an intercellular mechanism of lactate secretion. Interestingly, although brown fat expansion and enhanced functionality may enhance systematic energy expenditure and affect other metabolic tissues, such as white adipose tissues and livers, we found that FBXW7 modulation also regulated S6K1 in iWAT and beige adipocytes, suggesting the general existence of the regulatory axis in other fat depots. Therefore, our study provides crucial mechanism for FBXW7 in the regulation of thermogenic fat biology.

FBXW7, as a critical tumor suppressor, targets various substrates for proteasome-mediated degradation of oncoproteins, including cyclin E, c-Myc, Mcl-1, mTOR, Jun, and Notch (Yeh et al, 2018). Notably, recent studies highlighted FBXW7's function in the

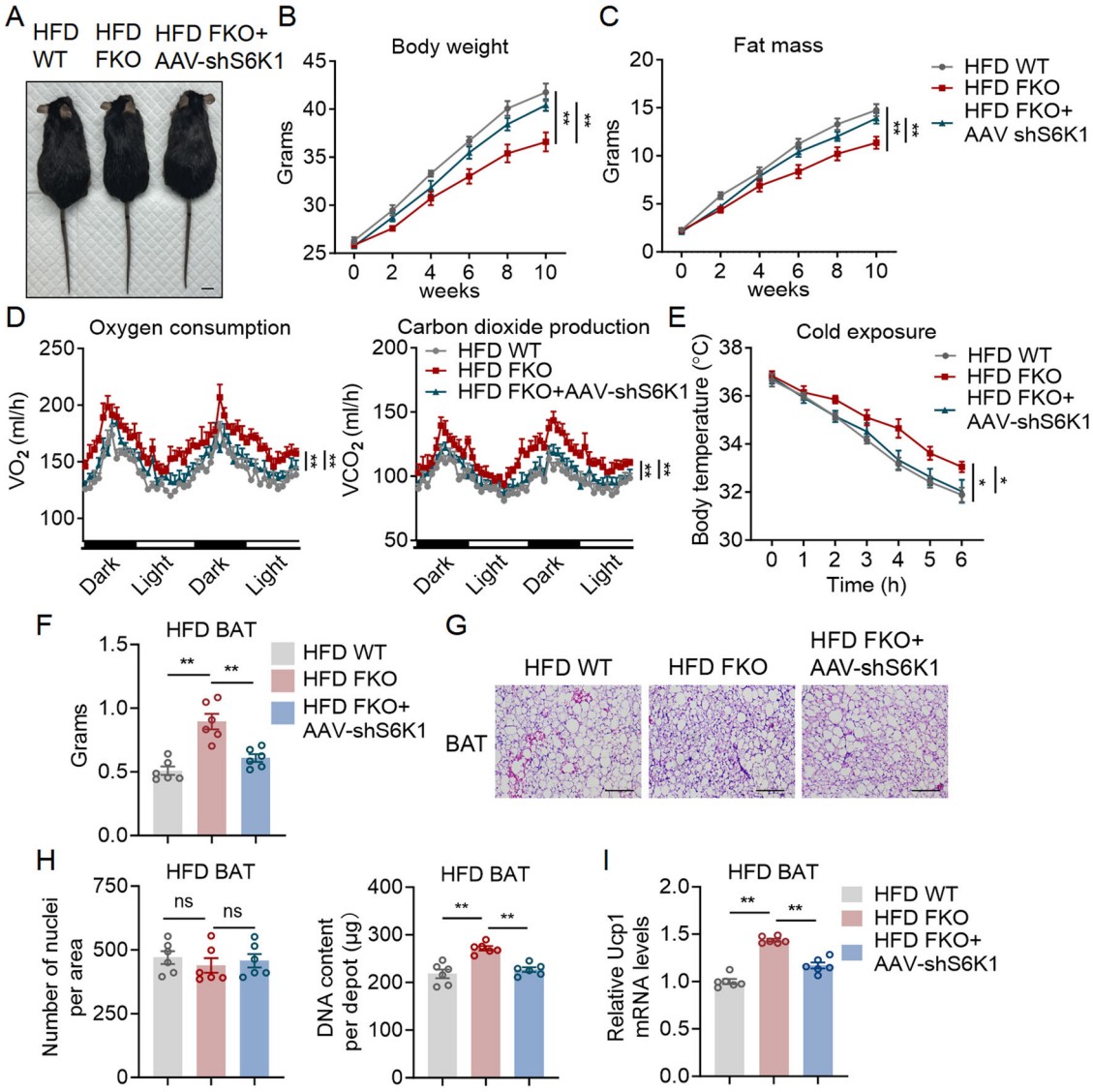

**Figure 7. S6K1 acts downstream of FBXW7 for the regulation of thermogenic fat function and obesity.**

(A–I) Metabolic performances of high fat diet fed WT, FBXW7-FKO mice and FBXW7 FKO + AAV-shS6K1 mice. AAV-shS6K1 and control AAV were injected both in BAT and iWAT of indicated mice and mice were fed with HFD. (A–C) Morphological appearance, body weight and fat mass; scale bar represents 1 cm, $n = 6$. (D) Energy expenditure as shown by oxygen consumption and carbon dioxide production of mice at an ambient temperature of 22 °C, $n = 5$. (E) Rectal temperatures of mice during 6 h cold exposure, $n = 6$. (F) Weights of BAT, $n = 6$. (G) Representative images of H&E staining of BAT, scale bar represents 100 μm. (H) Nuclei densities and genomic DNA content in BAT, $n = 6$. (I) Relative mRNA level of Ucp1 in BAT of WT, FBXW7 FKO mice and FBXW7 FKO + AAV-shS6K1 mice, $n = 6$. Data information: All data are representative of three individual experiments. $n$ refers to biological replicates. (B–E) Data are presented as mean ± SEM. Two-way ANOVA followed by Tukey's multiple comparison test was conducted and statistical significance denoted as *$p < 0.05$ and **$p < 0.01$ (B: $p = 0.0004$; C: $p = 0.0058$; D: Left panel: $p < 0.0001$; D: Right panel: $p < 0.0001$; E: $p = 0.0235$). (F, H, I) Data are presented as mean ± SEM. One-way ANOVA followed by Dunnett's multiple comparison test was conducted and statistical significance denoted as **$p < 0.01$. ns: not significant (F: $p < 0.0001$; H: Right panel $p < 0.0001$; I: $p < 0.0001$). Source data are available online for this figure.

metabolic field. For example, it has been found that hepatic FBXW7 regulated whole body lipid and glucose levels by targeting REV-ERBα or fetuin-A expression (Zhao et al, 2018; Zhao et al, 2016). As studies on post-translational regulation in fat metabolism are scarce, in this study, we focused on FBXW7's regulation on fat function and energy mechanism. Interestingly, the most striking phenotype of FBXW7-FKO mice is largely increased BAT mass with more DNA content per depot, indicative of enhance brown adipocyte expansion. Interestingly, this is similar to the phenotype

of mice with adipose tissue deficiency of LKB1, which consequently inhibited AMPK and activated mTOR-S6K1 signaling (Shan et al, 2016a, 2016b). mTORC1-S6K1 is a well-established regulator for cellular growth (Magnuson et al, 2012). Previous reports showed that FBXW7 is involved in the ubiquitination and degradation of mTOR in cancer cells (Mao et al, 2008), while in the current study, we examined major components of the mTOR signaling via mass spectrum and found that FBXW7 specifically targeted S6K1 in BAT, suggesting FBXW7 targets feature tissue selectivity for

ubiquitination. However, though we demonstrated that S6K1 was at least the direct target and central node for metabolic effects of FBXW7 modulation in BAT of mice, we can't fully exclude the possibility that other factors may also be targeted by FBXW7 and affected S6K1 activity. Future studies are warranted to examine FBXW7 targets in adipose tissues and other metabolic tissues.

It has to be noted that via adipose tissue transgenic and knockout mice models, we found that BAT mass was increased in both models. BAT mass is determined by both brown adipocyte numbers and cellular deposition, which underlies this seemingly contradictory phenotype. Indeed, detailed analysis revealed that in FBXW7-FTG mice, DNA content per depot was reduced and lipid deposition was increased in BAT, suggesting that the increase in BAT mass was due to reduced brown adipocytes numbers, which may cause functional decline and consequently cause lipid overload in BAT. Consistently, we observed decreased energy expenditure in these mice. On the other hand, FBXW7-FKO mice exhibited similar lipid deposition in BAT compared to WT mice, while DNA content per depot was increased, suggesting the larger BAT depots in these mice were due to enhanced brown adipocyte expansion.

In addition, it has been reported that FBXW7 controlled adipocyte differentiation by targeting C/EBPalpha for degradation (Bengoechea-Alonso and Ericsson, 2008). While in our present study, FBXW7 was specifically overexpressed or knocked out with the adiponectin promoter-driven Cre. In the fat tissues of mice, adiponectin only expressed in mature adipocytes but not in preadiopcytes. Therefore, in our system, FBXW7 was altered only in mature adipocytes, which may exempt its impact on preadipocytes and adipogenesis. Consistently, we further demonstrated that glycolysis was enhanced in BAT of FBXW7-FKO mice and lactate, the end product of glycolysis, mediated the crosstalk between mature adipocytes and preadipocytes to facilitate preadipocyte proliferation in KO mice. Lactate has been reported to promote cellular proliferation in multiple cell types including cancer cells, inflammatory cells and stem cells via its receptor GPR81 and downstream signaling transduction, or through epigenetic regulation of histone lactylation for broader transcriptional activation (Brown and Ganapathy, 2020). As FBXW7 is overexpressed or knocked out in mature adipocytes in our Adiponectin-cre mice models, the changes in brown preadipocyte proliferation and expansion are mediated by intercellular crosstalk. It is worthwhile to study whether FBXW7 impacts other secretion factors in mature adipocytes and their functions in the future.

The activation of mTOR signaling pathway promotes adipogenesis, while its inactivation reduces fat mass (Shan et al, 2016a, 2016b; Labbé et al, 2016). S6K1 has been shown to integrate nutrients and insulin signals, since S6K1 reduction and inhibition has been shown to cause glucose intolerance and insulin resistance (Pende et al, 2000). However, other studies have revealed that S6K1-deficient mice on a high fat diet were more insulin sensitive, due to the loss of a negative feedback loop, whereas S6K1 phosphorylates insulin receptor substrate 1 (IRS1) on its insulin resistance phosphorylation site at S307 and S636/S639 in fat, liver and muscle to inhibit insulin signaling (Um et al, 2004). These results suggested the complexity of S6K1 in the control of insulin sensitivity. In this study, we found that fat FBXW7 overexpression in mice showed reduced S6K1 and p-S6K1 levels in BAT and iWAT, accompanied with impaired insulin resistance. However, though FBXW7-FKO mice were much leaner than WT mice with

increased S6K1 in BAT, the insulin sensitivity was not altered, possibly due to the compensation effects of other FBXW7 targets. It seems that FBXW7 deficiency in mice manifested stronger phenotypes in lipid metabolism, including reduced white adipose tissue mass and fatty liver, which may underline the beneficial effects of FBXW7 knockout under chronic nutritional stresses. The mechanism of FBXW7 in the regulation of glucose metabolism through ubiquitin degradation needs to be further studied.

In summary, our data demonstrate that FBXW7 is a key regulator of S6K1 expression in brown fat and consequently regulates brown fat expansion and functionality via mTOR signaling mediated glycolysis and end product lactate mediated crosstalk between mature adipocytes and preadipocytes. This regulatory mechanism is also at least partially applied in iWAT and beige adipocytes. This study provides rational basis to suppress FBXW7 in adipose tissue for treating obesity and metabolic dysfunction in clinic.

# Methods

**Reagents and tools table**

| Reagent/Resource | Reference or Source | Identifier or Catalog Number |
|---|---|---|
| **Experimental models** | | |
| C57BL/6 (M. musculus) | Shanghai Research Center for Model Organisms | NO. SM-001 |
| db/db (M. musculus) | Shanghai Research Center for Model Organisms | NO. NM-KO-190663 |
| ob/ob (M. musculus) | Shanghai Research Center for Model Organisms | NO. NM-KO-00034 |
| CAG-loxP-Stop-loxP -EGE-FBXW7-WPRE-pA (M. musculus) | BIOCYTOGEN | NO. EGE-GJ-100 |
| Adiponectin-Cre (M. musculus) | Jackson | 01080 |
| FBXW7 LoxP (M. musculus) | Jackson | 017563 |
| Immortal brown preadipocytes (M. musculus) | SUNNCELL | SNP-M209 |
| Immortal beige preadipocytes (M. musculus) | SUNNCELL | SNP-M208 |
| 293T cells (H. sapiens) | ATCC | CRL-1573 |
| Escherichia coli DH5α | ThermoFisher | EC0112 |
| Escherichia coli BL21 | ThermoFisher | EC0114 |
| **Recombinant DNA** | | |
| pcDNA3.1(+) | ThermoFisher | V79020 |
| PcDNA3.1(+)-N-terminal Flag-tagged-FBXW7 | This study | N/A |
| PcDNA3.1(+)-HA-tagged S6K1 | This study | N/A |
| pET-28a(+) | Yeasen | 11905ES03 |
| pET-28a-His-tagged- FBXW7 | This study | N/A |
| pET-28a-HA-tagged- S6K1 | This study | N/A |
| Adenoviral-GFP | Hanbio biotechnology | N/A |

| Reagent/Resource | Reference or Source | Identifier or Catalog Number |
|---|---|---|
| Adenoviral-FBXW7 | Hanbio biotechnology | N/A |
| AdipoQ promoter driven AAV-shS6K1-adeno-associated virus | OBiO Technology | N/A |
| AdipoQ promoter driven AAV-GFP-adeno-associated virus | OBiO Technology | N/A |
| AdipoQ promoter driven AAV-shFBXW7-adeno-associated virus | OBiO Technology | N/A |
| **Antibodies** | | |
| Rabbit anti-FLAG-tag | Cell Signaling Technology | 87537 |
| Rabbit anti-HA-tag | Cell Signaling Technology | 3724 |
| Rabbit anti-FBXW7 | Abcam | ab192328 |
| Rabbit anti-pho-S6K1 (Ser371) | Cell Signaling Technology | 9208 |
| Rabbit anti-S6K1 | Cell Signaling Technology | 9202 |
| Rabbit anti-p-IRS-1 (Ser307) | Cell Signaling Technology | 2381 |
| Rabbit anti-IRS-1 | Cell Signaling Technology | 2382 |
| Rabbit anti-p-S6 Ribosomal Protein (RPS6)-S240/244 | Cell Signaling Technology | 5364 |
| Rabbit anti-p-PI3K p85 (Tyr458) | Cell Signaling Technology | 4228 |
| Rabbit anti-PI3K p85 | Cell Signaling Technology | 4292 |
| Rabbit anti-p-AKT (Ser473) | Cell Signaling Technology | 4060 |
| Rabbit anti-AKT | Abcam | ab314110 |
| Rabbit anti-p-GSK3β (Ser9) | Cell Signaling Technology | 9323 |
| Rabbit anti-GSK3β | Cell Signaling Technology | 9315 |
| Rabbit anti-UCP1 | Abcam | ab10983 |
| Mouse anti-β-Actin | Santa Biotechnology | sc-47778 |
| Protein A/G beads | Proteintech | PR40025 |
| APC/Cyanine7 anti-mouse CD31 | Biolegend | 102509 |
| APC/Cyanine7 anti-mouse CD45 | Biolegend | 103116 |
| PE anti-mouse Ly-6A/E (Sca-1) | Invitrogen | 108107 |
| **Oligonucleotides and other sequence-based reagents** | | |
| PCR primers | This study | Appendix Table S2 |
| S6K1 siRNA | This study | N/A |
| **Chemicals, Enzymes and other reagents** | | |
| Insulin | Sigma-Aldrich | I9278 |
| D-glucose | Sigma-Aldrich | G8270 |
| PureLink™ Genomic DNA Extraction Kit | Invitrogen | K182001 |
| RNAiso Plus | Takara | 9108 |

| Reagent/Resource | Reference or Source | Identifier or Catalog Number |
|---|---|---|
| PrimeScriptTM RT Master Mix | TaKaRa | RR036A |
| Hieff qPCR SYBR Green Master Mix | Yeasen | 11202ES03 |
| RIPA lysis buffer | Thermo Fisher | 89900 |
| Phosphatase inhibitor cocktail | Thermo Fisher | 78442 |
| PMSF | Thermo Fisher | 36978 |
| Lipofectamine 2000 Transfection Reagent | Thermo Fisher | 11668030 |
| NC membranes | PALL | 66485 |
| Bovine serum albumin | Sigma-Aldrich | V900933 |
| Dulbecco's Modified Eagle Medium | Gibco | 11995065 |
| Fetal bovine serum | Gibco | 10270 |
| Penicillin/streptomycin | Gibco | 15070063 |
| Dexamethasone | Sigma-Aldrich | D1756 |
| Triiodothyronine | Sigma-Aldrich | 642511 |
| Indomethacin | Sigma-Aldrich | I7378 |
| 3-isobutyl-1-methylxanthine | Sigma-Aldrich | I5879 |
| Rosiglitazone | Sigma-Aldrich | R2408 |
| 10% neutral formalin | Sigma-Aldrich | HT501128 |
| Collagenase type II | Sigma-Aldrich | C6885 |
| Hematoxylin and eosin staining | Beyotime | C0105S |
| Total triglyceride assay kit | BioVision | K622 |
| Total cholesterol assay kit | Sigma-Aldrich | MAK043 |
| High-density lipoprotein-cholesterol assay kit | Sigma-Aldrich | MAK045 |
| Low-density lipoprotein-cholesterol assay kit | Sigma-Aldrich | MAK045 |
| L-Lactate assay kit | Abcam | ab65331 |
| BCA Protein Assay Kit | Beyotime | P0012S |
| Silver staining kit | Sangon Biotech | C500021 |
| BeyoClick™ EdU-488 Cell Proliferation Assay Kit | Beyotime | C0071S |
| Cell counting kit-8 (CCK-8) | Beyotime | C0037 |
| **Software** | | |
| MaxQuant software | https://www.maxquant.org/ | |
| FlowJo v.10 | https://www.bdbiosciences.com/en-us/products/software/flowjo-v10-software | |
| GraphPad Prism | https://www.graphpad-prism.cn/ | |
| Image J | https://imagej.nih.gov/ij/index.html | |
| **Other** | | |
| High fat diet (60%) | Research Diet (D12492) | |
| AccuFat MRI system | MAG-MED AccuFat-1050 | |
| Blood glucose monitor | OneTouch Verio Flex® | |
| Comprehensive Lab Animal Monitoring System | Columbus Instruments | |
| Roche Light-Cycler480 | Roche | |

| Reagent/Resource | Reference or Source | Identifier or Catalog Number |
|---|---|---|
| Odyssey ®CLx Imaging System | LI-COR | |
| Light microscopy | Abaton Scan 300/Color scanner | |
| Q Exactive Mass Spectrometer | Thermo Scientific | |
| BD LSRFortessa™ Cell Analyzer | BD Biosciences | |
| Fluorescence microscope | Olympus Corporation | |
| SpectraMax microplate reader | Molecular Devices | |

## Animal studies

All animal studies were carried out following the guidelines approved by the Ethics Committee of Animal Experiments of East China Normal University (m20240104). We followed the ARRIVE guideline 2.0 (Percie du Sert et al, 2020). All necessary efforts were made to minimize the suffering of the animals. Male C57BL/6 (Shanghai Research Center for Model Organisms), db/db (Shanghai Research Center for Model Organisms), and ob/ob (Shanghai Research Center for Model Organisms) mice aged 8 weeks were purchased from Shanghai Research Center for Model Organisms. FBXW7-LoxP mice were generated by using the EGE system based on CRISPR/Cas9, and CAG-loxP-Stop-loxP-EGE-FBXW7-WPRE-pA was inserted at the Rosa26 site (BIO-CYTOGEN). Then FBXW7 fat-conditional overexpressed mice (FBXW7-FTG mice) was generated by crossing FBXW7-LoxP mice with Adiponectin-Cre mice. Mice with a targeted deletion of FBXW7 in adipose tissues (FBXW7-FKO) were generated by crossing the FBXW7 LoxP mice (Jackson Laboratory) with Adiponectin-Cre mice (Jackson Laboratory). We used littermate controls in all animal experiments ($n = 5$ for metabolic chamber and $n = 6$ for other experiments). Mice were randomly divided and free access to food and water with a 12 h/ 12 h light/dark cycle at an ambient temperature of 22 °C. For diet studies, 8-week-old mice fed with chow diet or high fat diet (60%, ResearchDiet) for indicated time. For monitoring metabolic parameters, body weight and body compositions were measured every two weeks using AccuFat MRI system (AccuFat-1050, MAG-MED). Oxygen consumption ($VO_2$), carbon dioxide production ($VCO_2$), energy expenditure (Heat), food intake and locomotor activity were measured by Comprehensive Lab Animal Monitoring System (CLAMS, Columbus Instruments). For CL-316,243 treatment, oxygen consumption was calculated to ml of $O_2 \cdot h^{-1}$ (Sveidahl Johansen et al, 2021) or ml of $\Delta O_2 \cdot min^{-1} \cdot (kg$ of body weight$^{-0.75})$ (Golozoubova et al, 2006). For cold exposure, mice were individually caged without beddings and exposed to 4 °C for 6 h. Rectal temperatures were measured each hour (Physitemp Instruments). For the insulin tolerance test (ITT), mice were injected with insulin in saline (1.25 U/kg body weight) intraperitoneally. For the glucose tolerance test (GTT), mice were fasted overnight and injected with D-glucose in saline (1.25 g/kg) intraperitoneally. Plasma glucose levels were measured at 0, 15, 30, 60, 90, and 120 min after injection with an blood glucose monitor (Bayer).

## Genomic DNA extraction and real-time PCR

Genomic DNA of tissues was extracted using PureLink™ Genomic DNA Extraction Kit (Invitrogen) according to the manufacturer's instructions. Total RNA was isolated from tissues and cells using

RNAiso Plus (Takara) and 1 μg of RNA was reversed using the PrimeScript™ RT Master Mix (TaKaRa) according to the manufacturer's instructions. Quantitative real-time PCR was performed using the Hieff qPCR SYBR Green Master Mix (Yeasen) on the Roche Light-Cycler480 (Roche). Relative mRNA levels were calculated using the 2-ΔΔCt method and sequences of qPCR primers were listed in Appendix Table S2.

## Immunoprecipitation and protein analysis

Total protein from tissues or cultured cells was extracted by the RIPA lysis buffer (Thermo Fisher) containing protease and phosphatase inhibitor cocktail (Thermo Fisher) and PMSF (Thermo Fisher). The equal amounts of total protein lysates were loaded into 10% SDS-PAGE and transferred to NC membranes (PALL). Membranes were blocked in 5% bovine serum albumin (BSA) (Sigma-Aldrich) for 2 h and incubated with primary antibodies overnight at 4 °C, including the following antibodies: anti-FLAG-tag antibody (Cell Signaling Technology), anti-HA-tag antibody (Cell Signaling Technology), anti-FBXW7 antibody (Abcam), anti-pho-S6K1 (Ser371) antibody (Cell Signaling Technology), anti-S6K1 antibody (Cell Signaling Technology), anti-p-IRS-1 (Ser307) antibody (Cell Signaling Technology), anti-IRS-1 antibody (Cell Signaling Technology), anti-p-S6 Ribosomal Protein (RPS6)-S240/244 antibody (Cell Signaling Technology), anti-p-PI3K p85 (Tyr458) antibody (Cell Signaling Technology), anti-PI3K p85 antibody (Cell Signaling Technology), anti-p-AKT (Ser473) antibody (Cell Signaling Technology), anti-AKT antibody (Abcam), anti-p-GSK3β (Ser9) antibody (Cell Signaling Technology), anti-GSK3β antibody (Cell Signaling Technology), anti-UCP1 antibody (Abcam) and β-Actin (Santa Biotechnology). After washing with PBST, HRP-conjugated secondary antibodies were incubated for 1 h at RT. The results of western blot were performed using Odyssey ®CLx Imaging System (LI-COR).

## Human adipose tissue biopsy samples and human participants for genotyping

Human fat biopsies were obtained from the subcutaneous fat of control subjects (18 kg/m² < BMI < 25 kg/m²) and obese (BMI ≥ 30 kg/m²) subjects in the First Affiliated Hospital of Wenzhou Medical University with approval of Human Research Ethics Committee of the hospital (KY2021-173). For human FBXW7 genotyping analysis, subjects were included from Shanghai Nicheng Cohort Study. Detailed standard about recruitment, clinical data acquisition and genotyping methods were described previously (Li et al, 2022). Human study was approved by Shanghai Jiaotong University Affiliated Sixth People's Hospital according to Declaration of Helsinki II (2015-KY-002(T)). Written informed consent was obtained from all subjects. The metabolic traits of human participants were summarized in Appendix Table S1.

## Cell culture

Immortal brown (SUNNCELL) and beige preadipocytes (SUNN-CELL) were cultured in Dulbecco's modified Eagle medium (DMEM, Gibco) supplemented 20% fetal bovine serum (FBS, Gibco) and 1% penicillin/streptomycin (P/S, Gibco) at 37 °C with 5% $CO_2$. when reached confluent, cells were differentiated in

induction medium (10% DMEM with 10% FBS, 1% P/S, 1 μmol/L dexamethasone (Sigma-Aldrich), 1 nmol/L triiodothyronine (T3) (Sigma-Aldrich), 0.125 mmol/L indomethacin (Sigma-Aldrich), 0.5 mmol/L 3-isobutyl-1-methylxanthine (IBMX) (Sigma-Aldrich), 1 μg/mL insulin (Sigma-Aldrich) and 1 μmol/L rosiglitazone (Sigma-Aldrich). After 2 days' induction, cells were switched to the maintenance medium (10% DMEM with 10% FBS, 1% P/S, 0.5 mmol/L T3 and 1 μg/mL insulin) every 2 days.

Primary stromal vascular fractions (SVF) were isolated from mice brown fat and subcutaneous fat and digested with 2 mg/mL collagenase type II (Sigma-Aldrich) at 37 °C for 20 min (Ma et al, 2015). Cell suspensions were centrifuged, washed, resuspended and plated in wells. The isolated SVFs from BAT and iWAT were differentiated and cultured with medium containing 10% FBS, 1% P/S supplemented with 5 μg/mL insulin, 1 nmol/L T3, 1 μmol/L dexamethasone, 0.5 mmol/L indomethacin, and 1 μmol/L rosiglita-zone for 48 h and subsequently cultured in maintenance medium (5 μg/mL insulin and 1 nmol/L T3) every 2 days. Regular testing confirmed the absence of mycoplasma contamination in all cells.

## siRNA transfection

siRNA was designed and synthesized by GenePharma (Shanghai, China). siS6K1 sequence: Sense (5'-3'): CCUGUCAGCCCAGU-CAAAUTT and Antisense (5'-3'): AUUUGACUGGGCUGA-CAGGTT. siRNA was transfected into differentiated brown adipocytes using Lipofectamine 2000 Transfection Reagent (Thermo Fisher) according to the manufacturer's protocol.

## Histological analysis

Adipose and liver tissues were dissected and fixed in 10% neutral formalin (Sigma-Aldrich). The fixed tissues were dehydrated and embedded in molten paraffin wax. The paraffin blocks were cut into 5 μm sections and were subjected to hematoxylin and eosin staining (Beyotime) according to the manufacturer's instructions. The image were acquired by light microscopy (Abaton Scan 300/Color scanner). The adipocyte sizes were quantified by ImageJ.

## Serum parameters, liver triglyceride, and lactate level determination

Serum parameters including total triglyceride (TG, BioVision), total cholesterol (TC, Sigma-Aldrich), high-density lipoprotein-cholesterol (HDL-C, Sigma-Aldrich) and low-density lipoprotein-cholesterol (LDL-C, Sigma-Aldrich) levels were measured via colorimetric assays with commercially available kits according to the manufacturer's instructions. Liver triglyceride was extracted using 5% NP-40 and heated at 90 °C for twice. The transparent supernatant was collected after centrifuging at 12,000 rpm for 2 min. TG measured in liver extracts were further normalized to protein concentration. The lactate levels in tissues or cell culture media were measured using L-Lactate assay kit (Abcam) according to the manufacturer's instructions.

## Tissue proteome extraction, digestion, and 6-plex TMT labeling

The tissues were lysed with lysis buffer (8 M urea, 100 mM $NH_4HCO_3$, and protease inhibitor 10% (v/v)). The lysis was further sonicated by for 5 min (2 s on and 5 s off) with 30% energy on the ice. The protein concentration was determined by using BCA assay. Then the protein-cysteine residues were reduced with 5 mM dithiothreitol (DTT) at 56 °C for 30 min, and alkylated with iodoacetamide (IAA) in the dark at room temperature for 30 min. The alkylation was finally quenched by add another 5 mM DTT and incubate 30 min at room temperature. Each protein solution was subjected to a 4-fold dilution with 100 mM $NH_4HCO_3$ and digested by sequence grade trypsin enzyme (Hualishi scientific) at a trypsin-to-protein ratio of 1:50 (w/w). The trypsin digestion was incubated at 37 °C for 16 h. Trypsin was further added at trypsin-to-protein ratio of 1:100 (w/w) and incubated at 37 °C for another 4 h to make sure the proteins were completely digested. The tryptic peptides were desalted through SepPak $C_{18}$ cartridges and vacuum-dried before 6-plex TMT labeling. The TMT labeling was according to the manufacturer's protocol.

## Offline-HPLC fractionation, LC-MS/MS analysis, and database searching

The 6-plex TMT-labeled peptides were separated by off-line high pH reverse-phase HPLC with an 80-min gradient. The elution of each minute was collected and finally combined into 20 fractions. The elution fraction was vacuum-dried before mass spectrometric analysis. All the dried samples were dissolved in buffer A (0.1% formic acid (FA) in water, v/v) and then detected by Easy-nanoLC 1000 system tandem with Orbitrap Fusion mass spectrometer with 60 min gradient. The peptides were separated by a home-made capillary analysis column (75 μm ID × 25 cm) with packing $C_{18}$ resins (3 μm particle size, 120 Å pole size, Dr. Maisch, USA). The gradient was as follows: 0–35 min, 5–30% buffer B (0.1% FA in 90% ACN); 35–50 min, 30–45% buffer B; 50–55 min, 45–80% buffer B; 55–60 min, 80% buffer B. For the MS1, the ions were detected by orbitrap analyzer with resolution set to 60,000 (m/z 200), the scan range was set as m/z 450 to 1500, the automatic gain control (AGC) targets were set to 1,000,000 and the maximum injection time was 50 ms. For MS/MS scan, the top 10 abundant precursor ions were fragmented under high collision dissociation (HCD) with normalized collision energy of 40%. The fragment ions were detected by orbitrap analyzer with resolution set to 15,000 (m/z 200), the maximum injection time was set to 90 ms, the AGC targets were set to 500,000. All the mass spectrometry data were searched against uniprot-Mouse proteome database (version 20170410) through MaxQuant software (version 1.4.1.2). Enzyme specificity was set to trypsin. The 6-plex TMT label on lysine and peptide N-termini and carbamidomethylation of cysteine were set as fixed modifications. The acetylation of protein N-term and oxidation of methionine were set as variable modifications. The maximum missed cleavages were set at 2. The false discovery rate (FDR) at both protein and peptide levels were filtered with 1%.

## Plasmids construction, adenoviruses, and adeno-associated virus treatment

The pcDNA3.1 vector was used to construct N-terminal Flag-tagged full-length FBXW7 and HA-tagged S6K1 sequences and transfected into *Escherichia coli* DH5α (ThermoFisher) for plasmids construction. The pet28a vector was used to construct His-tagged full-length FBXW7 and HA-tagged S6K1 sequences and

transfected into *Escherichia coli* BL21 (ThermoFisher) for plasmids construction. The adenoviral vector expressing GFP or FBXW7 was obtained from Hanbio biotechnology (Shanghai, China). The adeno-associated virus carrying adipoQ promoter driven AAV-shS6K1 and control was obtained from OBiO Technology (Shanghai, China). The adeno-associated virus carrying adipoQ promoter driven shS6K1 or control was injected locally in both BAT and iWAT at the concentration of $2 \times 10^{10}$ vg (vector genome) in a final volume of 50 μL to ensure direct action on both thermogenic fats. For BAT, the surgery was performed by making a small incision on the back of each mouse's scapula area under anesthesia to obtain a clearer view and injected BAT bilaterally with 50 μL AAV-CON or AAV-shS6K1. For iWAT, the skin was disinfected and bilateral inguinal fats were injected subcutaneously with 50 μL AAV-CON or AAV-shS6K1. The adeno-associated virus carrying adipoQ promoter driven shFBXW7 or control was injected locally in BAT at the concentration of $5 \times 10^{10}$ vg (vector genome) in a final volume of 50 μL to ensure direct action on BAT. Brown or beige adipocytes were differentiated and then infected with adenovirus for overexpression of GFP or FBXW7 for 2 days. Inguinal fat pads were injected with ADV-GFP or ADV-FBXW7 for 4 days. BAT and iWAT of FBXW7-FKO and WT mice were injected with AAV-shS6K1 or control for 12 weeks.

## Immunoprecipitation-mass spectrum

Mature brown adipocytes were infected with ADV-FBXW7 for 48 h before collecting the protein lysate. Cells were lysed by IP lysis buffer and the supernatant was mixed with anti-FBXW7, anti-IgG and protein A/G beads overnight at 4 °C. The beads were washed by IP washing buffer and the immunprecipitates were analyzed by SDS-PAGE and silver staining (Sangon Biotech). The proteins were then analyzed by high-performance liquid chromatography mass spectrum and a Q Exactive Mass Spectrometer (Thermo Scientific, Waltham, Massachusetts) at Shanghai Applied Protein Technology. The original files were transformed with Proteomics Tools 3.1.6 software, and Proteome Discoverer 2.5 software was used for database screening.

## Co-immunoprecipitation and ubiquitination assay

293T cells were collected in IP lysis buffer containing 25 mM Tris-HCl, 150 mM NaCl, and 1% NP-40 supplemented with 1 μM protease inhibitor, 1 μM phosphatase inhibitor, 1 μM DTT, and 2.5 mM PMSF. Lysates were sonicated for 5 min on ice and centrifuged at 12,000 rpm for 10 min at 4 °C. The supernatant was collected and subjected to co-immunoprecipitation (co-IP) or ubiquitination assay. For Co-IP, 1 μg of anti-Flag-tag antibody, anti-HA-tag antibody, or IgG and 30 μl of protein A/G beads were added to each sample in IP binding buffer for overnight incubation at 4 °C. On the second day, beads were washed three times using IP washing buffer and resuspended in 30 μL of 1× SDS sample buffer. Samples were boiled for 5 min and centrifuged before storage at −80 °C until use. For the ubiquitination assay, the cell lysates were incubated with anti-HA-tag antibody and protein A/G beads overnight at 4 °C. The beads were then washed three times with IP washing buffer and subjected to immunoblot analysis using anti-ubiquitin antibody. The immunoblot was performed using standard protocols.

In vitro ubiquitination assay was performed as previously described (Ji et al, 2021). Briefly, Recombinant His-FBXW7 protein, HA-S6K1 protein, E1, E2, ubiquitin, and an ATP-regeneration system were used. Samples were incubated at 30 °C for 90 min and analyzed by immunoblotting.

For endogenous co-immunoprecipitation, brown fat was extracted by IP lysis buffer and lysates were sonicated for 5 min on ice and centrifuged at 12,000 rpm for 10 min at 4 °C. The supernatant was collected and 1 μg of anti-FBXW7 antibody, anti-S6K1 antibody, or IgG and 30 μL of protein A/G beads were added to each sample in IP binding buffer for overnight incubation at 4 °C. On the second day, beads were washed three times using IP washing buffer and resuspended in 30 μL of 1× SDS sample buffer. Samples were boiled for 5 min and centrifuged before storage at −80 °C until use.

## Flow cytometry assay

SVF cells were centrifuged at $800 \times g$ for 5 min at 4 °C after lysis of red blood cell, and then resuspended in staining buffer (0.1% BSA in PBS), following with cell-surface antibodies incubation at 4 °C for 60 min, including APC/Cyanine7 anti-mouse CD31 (1:200; Biolegend), APC/Cyanine7 anti-mouse CD45 (1:100; Biolegend), PE anti-mouse Ly-6A/E (Sca-1) (1:100; Invitrogen). Flow cytometry data were collected using BD LSRFortessa™ Cell Analyzer (BD Biosciences). All data analysis was performed using the flow cytometry analysis program FlowJo v.10.

## Ethynyl deoxyuridine (EdU) staining

Brown preadipocytes were cultured in 48-well plate under 37 °C and 5% $CO_2$, and pretreated with conditioned medium or Lactate (0, 5, 10 μM) for 24 h. Cell Proliferation was assayed using BeyoClick™ EdU-488 Cell Proliferation Assay Kit (Beyotime) according to the manufacturer's instructions. These results were visualized by a fluorescence microscope (Olympus Corporation), and the signals were counted in random visional fields.

## Cell viability assay

Cell viability was assessed by the cell counting kit-8 (CCK-8) (Beyotime). Briefly, cells were incubated with CCK-8 solution at 37 °C for 2 h and the absorbance was measured at 450 nm with a SpectraMax microplate reader (Molecular Devices).

## Statistical analyses

The experiments were randomized, while the investigator was not blinded. All values specified in Figures were represented as mean ± SEM. Data were tested for normally distribution with the Shapiro–Wilk normality test. Student's unpaired two-tailed t tests were used to compare two sets of conditions and Welch ANOVA followed by Dunnett's post hoc test was used to compare multiple sets of conditions. A 95% confidence interval was used considering differences statistically significant when $P < 0.05$. ANCOVA with body weight as covariant was used to analyze oxygen consumption data by SPSS software (Müller et al, 2021). Statistical analyses were performed with PRISM software version 9.5.0 (www.graphpad.com). The $P$ values were designed as follows: *$P < 0.05$; **$P < 0.01$, ns: not significant. The number of samples used for each experiment was described in the

figure panels or corresponding figure legends. The synopsis image was created with BioRender.com.

## Data availability

This study includes no data deposited in external repositories.

The source data of this paper are collected in the following database record: biostudies:S-SCDT-10_1038-S44319-024-00337-w.

## Peer review information

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

## Acknowledgements

This project is supported by funds from National Natural Science Foundation of China (32325024, 82300979, 32222024, 32271224, 32071148, 22225702, 82000802), National Key Research and Development Program of China (2023YFA1800400, 2019YFA09004500), Science and Technology Commission of Shanghai Municipality (22ZR1421200, 21140904300), Natural Science Foundation of Chongqing, China (CSTB2022NSCQ-JQX0033), Key Research and Development Project of Zhejiang Province (2021C03069), Zhejiang Provincial Natural Science Foundation of China (LY20H070003), Innovative research team of high-level local universities in Shanghai (SHSMU-ZDCX20212700), Shanghai Research Center for Endocrine and Metabolic Diseases (2022ZZ01002) and Shanghai Municipal Key Clinical Specialty, China Postdoctoral Science Foundation (2023M741184), Fundamental research funds for the central universities, ECNU public platform for Innovation (011), Specific Pathogen Free Laboratory Animal Centre and the instruments sharing platform of School of Life Sciences.

## Author contributions

**Jian Yu**: Formal analysis; Investigation; Methodology; Writing—original draft. **Xuejiang Gu**: Formal analysis; Investigation; Methodology. **Yingying Guo**: Software; Formal analysis; Supervision; Methodology. **Mingyuan Gao**: Formal analysis; Investigation; Methodology. **Shimiao Cheng**: Software; Formal analysis; Investigation. **Meiyao Meng**: Validation; Investigation; Methodology. **Xiangdi Cui**: Resources; Software; Methodology. **Zhe Zhang**: Resources; Software; Methodology. **Wenxiu Guo**: Resources; Software; Methodology.

**Dandan Yan**: Software; Formal analysis; Validation. **Maozheng Sheng**: Resources; Software; Formal analysis. **Linhui Zhai**: Software; Formal analysis; Investigation. **Jing Ji**: Validation; Investigation; Methodology. **Xinhui Ma**: Resources; Formal analysis; Methodology. **Yu Li**: Resources; Software; Formal analysis. **Yuxiang Cao**: Formal analysis; Investigation; Methodology. **Xia Wu**: Formal analysis; Methodology. **Jiejie Zhao**: Formal analysis; Validation; Methodology. **Yepeng Hu**: Formal analysis; Validation; Methodology. **Minjia Tan**: Resources; Software. **Yan Lu**: Resources; Software; Validation. **Lingyan Xu**: Conceptualization; Supervision; Writing—original draft; Writing—review and editing. **Bin Liu**: Conceptualization; Resources; Writing—review and editing. **Cheng Hu**: Conceptualization; Supervision; Writing—review and editing. **Xinran Ma**: Conceptualization; Supervision; Writing—original draft; Writing—review and editing.

Source data underlying figure panels in this paper may have individual authorship assigned. Where available, figure panel/source data authorship is listed in the following database record: biostudies:S-SCDT-10_1038-S44319-024-00337-w.

## Disclosure and competing interests statement

The authors declare no competing interests.

