## [Peer Review File · EMBO Reports]

E3 Ligase FBXW7 Suppresses Brown Fat Expansion and Browning of White Fat

Jian Yu, Xuejiang Gu, Yingying Guo, Mingyuan Gao, Shimiao Cheng, Meiyao Meng, Xiangdi Cui, Zhe Zhang, Wenxiu Guo, Dandan Yan, Maozheng Sheng, Linhui Zhai, Jing Ji, Xinhui Ma, Yu Li, Yuxiang Cao, Xia Wu, Jiejie Zhao, Yepeng Hu, Minjia Tan, Yan Lu, Lingyan Xu, Bin Liu, Cheng Hu, and Xinran Ma

Corresponding author(s): Xinran Ma (xrma@bio.ecnu.edu.cn) , Cheng Hu (alfredhc@sjtu.edu.cn), Lingyan Xu (lyxu@bio.ecnu.edu.cn), Bin Liu (liubin@jou.edu.cn)

Review Timeline:

Submission Date:	8th Jul 23
Editorial Decision:	20th Jul 23
Appeal Received:	4th Feb 24
Editorial Decision:	15th Mar 24
Revision Received:	15th Jul 24
Editorial Decision:	15th Oct 24
Revision Received:	26th Oct 24
Accepted:	8th Nov 24

Editor: Deniz Senyilmaz Tiebe

Transaction Report:

Editor:

We appreciate your study demonstrating that E3 ligase FBXW7 suppresses brown fat thermogenesis by a mechanism involving S6K1 degradation. We realize that these findings are as such of interest to the field. However, we also note that, in our view, the regulatory link between FBXW7 and S6K1 has not been investigated in sufficient depth - i.e. was not supported with endogenous IP and in vitro ubiquitination assays. Moreover, the functional relevance of the proposed link between FBXW7 and S6K1 remains to be demonstrated. In light of these combined reservations, we concluded that the advance provided is not sufficient for EMBO Reports. That said, we would be happy to send the manuscript out for formal peer-review should you be willing to include additional data addressing these concerns.

Thanks for the editor's valuable comments. We have conducted the following experiments to strengthen the the regulatory link between FBXW7 and S6K1 thus largely improve our work based on the editor's suggestions.

Firstly, to examine whether S6K1 is a FBXW7 substrate, In addition to LC-MS/MS analysis on BAT from WT and FBXW7-FKO mice under HFD as we reported in the previous submission (Table EV2), in which we showed significantly increased ribosomal proteins in BAT of FBXW7-FKO mice under KEGG pathway analysis (Fig. 4A), we also performed immunoprecipitation coupled with mass spectrum (IP-MS) to identify the interacting proteins of FBXW7 in FBXW7 overexpressed brown adipocytes (Table EV3). Of note, venn diagram analysis of LC-MS/MS and IP-MS results suggested that S6K1 is a FBXW7 substrate (Fig 4B).

The new data has been added as Fig 4B.

Figure 4

In addition, based on the editor's suggestion, we have also performed *in vitro* ubiquitination assays. The results showed that recombinant FBXW7 protein interacts with S6K1 protein and mediates its ubiquitination *in vitro* (Fig 5A).

The new data has been added as Fig 5A.

Figure 5

In addition, the endogenous immunoprecipitation assay also showed that endogenous FBXW7 interacts with S6K1 in BAT (Fig 5C).

The new data has been added as Fig 5C.

Figure 5

Moreover, to further strengthen the functional relevance between FBXW7 and S6K1 in brown and beige fat function and energy metabolism *in vivo*, we performed rescue experiments by administrating AAV carrying adipoQ promoter driven shS6K1 construct (AAV-shS6K1) and control in BAT and iWAT of WT and FBXW7-FKO mice. The AAV-shS6K1 delivery achieved specific S6K1 reduction in BAT and iWAT in FBXW7-FKO mice and comparable levels in WT mice (Fig EV12A). When subjected to HFD feeding, FBXW7-FKO mice with fat-S6K1 knockdown showed increased body weights and fat mass, decreased energy expenditure and impaired thermogenic capacity after cold exposure, increased liver weights and lipid infiltration, as well as elevated serum lipid profiles compared to FBXW7-FKO mice, without changes in locomotor activity and food intake (Fig 7A-E and Fig EV12B-G). In addition, HFD fed FBXW7-FKO mice with fat-S6K1 knockdown showed significantly enlarged BAT depot with increased brown adipocyte sizes, more lipid deposition, while decreased numbers of nuclei per unit area (Fig 7F and G), genomic DNA content per depot (Fig 7H) and Ucp1 mRNA levels (Fig 12I), compared to those of FBXW7-FKO mice.

Meanwhile, the weights of iWAT and eWAT of HFD FBXW7-FKO mice with fat-S6K1 knockdown were enhanced with increased adipocyte sizes (Fig EV12H-J), decreased numbers of nuclei per unit area and genomic DNA content per depot, as well as reduced Ucp1 mRNA levels in iWAT, compared to those of FBXW7-FKO

mice (Fig EV12K and L). Taken together, these data suggested that S6K1 acts downstream of FBXW7 for the regulation of thermogenic fat function and obesity.

Figure 7

Figure S12

Dear Xinran,

Thank you for the submission of your research manuscript to our journal, which was now seen by three referees, whose reports are copied below.

Referees express interest in the proposed role of FBXW7 in regulation of adipose tissue metabolism. However, they also raise significant overlapping concerns that need to be addressed to consider publication here. In particular, referees find that additional analysis on BAT activity and energy expenditure (by ANCOVA method) is warranted. Moreover, referees point out issues regarding data presentation and statistical analyses.

Of note, while we agree with referee #3 that generation of BAT specific FBXO7 KO mice would strengthen the manuscript, we also find that it is not prerequisite for publication here.

Given these positive recommendations, we would like to invite you to submit a revised manuscript. Please revise your manuscript with the understanding that the referee concerns (as in their reports) must be fully addressed and their suggestions taken on board. Please address all referee concerns in a complete point-by-point response. Acceptance of the manuscript will depend on a positive outcome of a second round of review. It is EMBO reports policy to allow a single round of major experimental revision only and acceptance or rejection of the manuscript will therefore depend on the completeness of your responses included in the next, final version of the manuscript.

We realize that it is difficult to revise to a specific deadline. In the interest of protecting the conceptual advance provided by the work, we recommend a revision within 3 months. Please discuss the revision progress ahead of this time with me if you require more time to complete the revisions, or if you have questions or comments regarding the revision (also by video chat).

1. A data availability section providing access to data deposited in public databases is missing (where applicable).
2. Your manuscript contains statistics and error bars based on $n=2$. Please use scatter plots in these cases.

You can submit the revision either as a Scientific Report or as a Research Article. For Scientific Reports, the revised manuscript can contain up to 5 main figures and 5 Expanded View figures, and it should not exceed 27000 characters. If the revision leads to a manuscript with more than 5 main figures it will be published as a Research Article. In this case the Results and Discussion section should be separate. If a Scientific Report is submitted, these sections have to be combined. This will help to shorten the manuscript text by eliminating some redundancy that is inevitable when discussing the same experiments twice. In either case, all materials and methods should be included in the main manuscript file.

3) We replaced Supplementary Information with Expanded View (EV) Figures and Tables that are collapsible/expandable online. A maximum of 5 EV Figures can be typeset. EV Figures should be cited as 'Figure EV1, Figure EV2' etc... in the text and their respective legends should be included in the main text after the legends of regular figures.

4) a .docx formatted letter INCLUDING the reviewers' reports and your detailed point-by-point responses to their comments. As part of the EMBO publication's Transparent Editorial Process, EMBO reports publishes online a Review Process File (RPF) to

accompany accepted manuscripts. This File will be published in conjunction with your paper and will include the referee reports, your point-by-point response and all pertinent correspondence relating to the manuscript.

<https://www.embopress.org/page/journal/14693178/authorguide#transparentprocess>

5) a complete author checklist, which you can download from our author guidelines

<https://www.embopress.org/page/journal/14693178/authorguide>. Please insert information in the checklist that is also reflected in the manuscript. The completed author checklist will also be part of the RPF.

6) Please note that all corresponding authors are required to supply an ORCID ID for their name upon submission of a revised manuscript (<<https://orcid.org/>>). Please find instructions on how to link your ORCID ID to your account in our manuscript tracking system in our Author guidelines

<<https://www.embopress.org/page/journal/14693178/authorguide#authorshipguidelines>>

Additional information on source data and instruction on how to label the files are available:

<https://www.embopress.org/page/journal/14693178/authorguide#sourcedata>

9) Our journal encourages inclusion of *data citations in the reference list* to directly cite datasets that were re-used and obtained from public databases. Data citations in the article text are distinct from normal bibliographical citations and should directly link to the database records from which the data can be accessed. In the main text, data citations are formatted as follows: "Data ref: Smith et al, 2001" or "Data ref: NCBI Sequence Read Archive PRJNA342805, 2017". In the Reference list, data citations must be labeled with "[DATASET]". A data reference must provide the database name, accession number/identifiers and a resolvable link to the landing page from which the data can be accessed at the end of the reference. Further instructions are available at <http://www.embopress.org/page/journal/14693178/authorguide#referencesformat>

10) Regarding data quantification (see Figure Legends:

<https://www.embopress.org/page/journal/14693178/authorguide#figureformat>)

11) The journal requires a statement specifying whether or not authors have competing interests (defined as all potential or actual interests that could be perceived to influence the presentation or interpretation of an article). In case of competing

interests, this must be specified in your disclosure statement. Further information: <https://www.embopress.org/competing-interests>

12) Please also note our reference format:

I look forward to seeing a revised version of your manuscript when it is ready. Please let me know if you have questions or comments regarding the revision.

Kind regards,

Deniz

Deniz Senyilmaz Tiebe, PhD
Scientific Editor
EMBO Reports

Referee #1:

Yu et al. investigate the role of F-box and WD repeat domain-containing 7 (Fbxw7) within the adipose tissue of mice, by using multiple transgenic and dietary approaches. The authors first show how overexpressing and knocking out Fbxw7 negatively and positively affects the metabolism of the mice, respectively. While overexpression worsens the metabolic phenotype of the mice through suppressed energy expenditure and dysfunctional lipid metabolism, loss of Fbxw7 appears to improve the metabolism of mice and protect them from the deleterious effects of HFD. By utilizing Co-IP and in vitro ubiquitination assays, the authors elegantly show how Fbxw7 as part of an E3 ligase complex can bind to S6K1 on a molecular level, causing its subsequent degradation. Loss of Fbxw7 supposedly leads to elevated glycolysis and lactate production, increasing proliferation of brown adipocytes, which aids in the metabolic amelioration of the HFD. Furthermore, the authors nicely show that the metabolic phenotype seen with the loss of Fbxw7 can be reverted by silencing of S6K1. In conclusion, Yu and colleagues nicely demonstrate the important role of Fbxw7 within the adipose tissue of mice. The manuscript could be improved further through simple editing of the data, as well as some minor additional experiments. The fact that Fbxw7 binds to multiple proteins within the cell, as shown by the IP (53 binding partners), shows the complexity of its role as part of an E3-ligase. The major question is whether Fbxw7 only binds to S6K1 and if this alone results in the metabolic effects seen in this study, or if Fbxw7 also binds other important proteins, e.g. upstream of S6K1 with these effects only converging down on S6K1. For example, the authors already stated that Fbxw7 is a well-established tumor suppressor gene, which has also been shown to control adipocytes differentiation (PMID: 20534483). In the end, there is very little direct evidence that would support a role of Fbxw7 for brown fat expansion, non-shivering thermogenesis, and energy metabolism in the animal experiments. They are poorly controlled and the HFD feeding results in varying degrees of weight gain. This part of the manuscript needs major revisions.

Major points:

1. Body temperature measurement is not a good indicator of BAT activity, as it includes confounding changes in shivering activity and other cardiovascular physiological mechanisms. Please redo this experiment by subjecting the mice to a β 3-agonist such as CL316 243 and measure O₂ consumption as described in PMID: 21177944.
2. For investigating brown fat expansion (what precisely do the authors mean?), a systematic analysis of non-shivering thermogenesis in the animals is warranted,
3. Please use ANCOVA for determining if the changes in body weight are results of energy expenditure as described in PMID: 22205519.
4. How many mice were used for the experiments in all the figures? Were these littermate controls? When comparing Figures 3C and 7C, the curves of HFD FKO very much resemble each other. Was this a separate mouse experiment or were the previous data from the experiment in Figure 3 reused? It also appears odd that the authors would choose a different Y-axis labelling for Figure 7C. Please provide all details on statistics in the mouse experiments like repetitions, reproduction, mouse numbers.
5. It would be valuable to see western blots of mouse tissues, showing that overexpression and knockout of Fbxw7 were achieved on protein level. Similarly, I would like to see that the changes in mRNA expression of Ucp1 in the mice actually show the same effects on protein level.
6. Please always include all data points in the graphs, not just the bar with SD.

Minor points:

7. Figure 2 and 3 E: What's the temperature that the mice we subjected to here? 22{degree sign}C? 4{degree sign}C?
8. Please include proper positive controls in all validation experiments that utilize inhibitors (e.g. for proteasome inhibition with

MG-132, show increased ubiquitination)

9. Please attach all uncropped western blot pictures in the supplement.

10. The manuscript requires a grammar and spell check.

11. What about downstream targets of S6K1? For example S6RP? Is it also affected by the change in S6K1 protein levels? An additional Western blot here would be necessary. Furthermore, how is the insulin-signaling pathway influenced by overexpression and KO of Fbxw7?

12. Figures EV8I-M do not exist. EV20G also does not exist. Please review correct cross-referencing of your figures in the text.

13. Why are the ubiquitination and IP assays performed in HEK293T cells and not in brown adipocytes or another adipocyte cell line? The claimed effect of the authors appears to be in adipocytes.

14. How does the fat mass of HFD FKO and HFD FKO + shS6K1 change? Only the lean mass is shown. Although figure S12 H shows iWAT and eWAT weight, it does not show how the weights developed throughout the weeks of HFD feeding.

15. Fbxw7KO leads to increased proliferation through increased lactate production. Why don't the authors prove this by inhibiting the lactate transporter?

16. One of the targets of Fbxw7 according to the IP data is Pyruvate kinase PKM. If Fbxw7 normally degrades PKM via the UPS, knocking it down could be part of the reason why an increased glycolysis and lactate production was seen.

Referee #2:

The manuscript used human samples and different metabolic disease models and found that FBXW7 is potentially involved in disease progression. Thus, they demonstrated this via FBXW7 overexpression and knockout in the adipose tissue via various experiments. Finally, they pinpointed S6K1 as the substrate of FBXW7 via proteomics. This work is of novelty and physiological significance. However, there are a few concerns to be clarified and to make the work more convincing. Also, the author should correct writing mistakes carefully to make it more professional.

Major points

1. The authors used the FBXW7 antibody to examine its levels in different adipose depots of different models. As the work used FBXW7 TG and KO models, however, they have not verified the antibody in those models. To demonstrate regulation of FBXW8 during disease progression, the author should measure FBXW7 protein levels in different adipose depots and liver (corresponding to transcript levels in Fig1A). Moreover, FBXW7 mRNA levels have to be examined in HFD/Ob/Db models (as in Fig1C, 1D), which can provide more information about how FBXW7 is regulated.

2. What is dominant mechanism to mediate the effect of FBXW8 upregulation? cell autonomous suppression of UCP1 in brown and beige adipocytes or suppressed expansion? Are both mechanisms mediated by S6K1 regulation?

3. Can the author provide BMI index of Fig1E, as this might reflect obesity in better way?

4. Please re-analyse energy expenditure of all the related results in ANCOVA method as this is commonly recommended method for energy expenditure study.

5. It will make the thermogenesis study more convincing if the authors evaluate the related genes at protein levels, at least for UCP1. In the present version, only mRNA levels are provided.

6. Can the authors explain why the glucose and insulin sensitivity remained largely the same in the FBXW7 KO mice in the HFD feeding conditions (Fig 3D, 3E), considering that the body weight gain is significantly reduced after FBXW7 depletion?

7. According to the results, FBXW7 overexpression in the brown or beige adipocytes should decrease S6K1 and the lactate release to reduce preadipocyte expansion. The authors should measure lactate levels in both the in vitro and in vivo models after FBXW8 overexpression, just like FBXW8KO, since FBXW8 upregulation is claimed by the authors to aggravate obesity.

8. Please carefully describe in the method part how the authors knock down S6K1 in the iBAT and iWAT together via AAV, including the size of the promoter, the injection method as this might be technically challenging to achieve it.

9. Could the author include one group with only S6K1 knockdown in the adipose tissue?

Minor points:

1. Fig1C, 1D, FigS1A, B, the authors mis-spelled relative as relative. FigS3D, S7D, S9D, Percentage instead of percent might be better.

2. Thermogenic fat, including brown (BAT) and beige adipose tissues generate heat by decoupling proton gradient from oxidative phosphorylation and release it through uncoupling protein (UCP1) in mitochondrial membrane. Release should be 'releasing'.

3. thus are tightly involved in the adipose tissue functionality and onset of obesity. should be, Thus, they are tightly....

4. To evaluate the functions of FBXW7 in fat biology, we examined Fbxw7 expression levels in various metabolic organs and found that adipose tissues including BAT, inguinal WAT (iWAT) and epididymal WAT (eWAT), should be 'epididymal'.

5. Interestingly, we also observed increased numbers of nuclei per unit area and genomic DNA content per depot, as well as reduced Ucp1 expressions in iWAT of FBXW7-FTG mice. Based on FigS5L, nuclei and genomic DNA should be decreased.

6. these data suggests that FBXW7 aggravates obesity and metabolic dysfunctions in mice, possibly due to impaired tissue expansion and thermogenesis in thermogenic fat. It should be 'suggest'.

7. The thermogenic capacity of BAT of FBXW7-FTG mice were suppressed as evident by the decreased UCP1 levels (Fig EV4D). This sentence requires rephrasing.

8. Detailed analysis showed that as expected and opposite to FBXW7-FTG's phenotypes, Strikingly, we found that FBXW7-FKO

mice had significantly larger BAT compared to WT (Fig EV8A). This sentence requires rephrasing.

9. Indeed, recombinant FBXW7 protein mediated S6K1 protein ubiquitination in vitro (Fig 5A) and coimmunoprecipitation assays showed that FBXW7 physically interacted with S6K1 in overexpressed HEK29T and endogenously in BAT (Fig 5B and C). HEK29T should be 'HEK293T'.

10. Since we found that FBXW7 deficiency increased brown fat sizes and functions by impairing adipocyte expansion and thermogenesis, to further understand the intrinsic mechanisms, we performed flow.... This is a wrong statement according to the results.

There are so many grammar or wording mistakes as what I pointed out above. I recommend the authors to seek assistance from ones with better language skills.

Referee #3:

This manuscript examined the physiological role of a brown fat abundant E3 ubiquitin protein ligase FBXW7 in thermogenesis and energy homeostasis. Through fat specific overexpression or loss-of-function analysis, the authors concluded that FBXW7 targets S6K1 for ubiquitination and degradation, which induced lactate production. Lactate then act on preadipocytes to promote proliferation and differentiation. Overall, the research is interesting. However, conclusions and molecular mechanisms are needed to be further polished. In addition, the authors need to reformat the manuscript, like Fig EV means figure supplement or? It's hard to follow in the current format. More specifically, I have the following points need to be clarified by the authors.

1. How about brown fat specific FBXW7 KO mice, if the authors conclude the phenotype is due to thermogenesis and BAT manifested a very strong phenotype.
2. The authors concluded that FBXW7 expression is negatively correlated with thermogenic fat functionality at the beginning of their story. However, this conclusion is only based on the expression of FBXW7 in brown fat under different conditions such as HFD and in obese animals models (ob/ob and db/db). This conclusion is too farfetched. To consolidate this conclusion, the authors should at least preform a correlation analysis with UCP1 expression.
3. Please clarify how the Adipose-specific overexpression of FBXW7 is achieved? With adiponectin-cre, UCP1-cre, PDGFRA-cre or Cre with ERT2? Please check UCP1 expression at protein level. To conclude brown fat function is affected, the authors need to perform respirometry experiment with acute injection of NE or CL. As brown fat is enlarged, it would be necessary to calculate the whole brown fat level of UCP1 both at protein level and mRNA level.
4. Similar to what were observed in CD mice.... what are CD mice?
- 5 please rewrite the sentence "Detailed analysis showed that as expected and opposite to FBXW7-FTG's phenotypes, Strikingly, we found that FBXW7-FKO mice had significantly larger BAT compared to WT (Fig EV8A)."
6. "Since we found that FBXW7 deficiency increased brown fat sizes and functions by impairing adipocyte expansion and thermogenesis," how could this possible? brown fat function is thermogenesis or?
7. How does loss of FBXW7 lead to increase of glycolysis or mTOR signaling? Whats the role of S6K1 here? Are glycolysis genes are decreased in FBXW7 overexpressed animals?
8. There is no UCP1 protein expression analysis in the whole paper. No brown fat functional analysis like NE or CL test analysis.

Referee #1:

Yu et al. investigate the role of F-box and WD repeat domain-containing 7 (Fbxw7) within the adipose tissue of mice, by using multiple transgenic and dietary approaches. The authors first show how overexpressing and knocking out Fbxw7 negatively and positively affects the metabolism of the mice, respectively. While overexpression worsens the metabolic phenotype of the mice through suppressed energy expenditure and dysfunctional lipid metabolism, loss of Fbxw7 appears to improve the metabolism of mice and protect them from the deleterious effects of HFD. By utilizing Co-IP and in vitro ubiquitination assays, the authors elegantly show how Fbxw7 as part of an E3 ligase complex can bind to S6K1 on a molecular level, causing its subsequent degradation. Loss of Fbxw7 supposedly leads to elevated glycolysis and lactate production, increasing proliferation of brown adipocytes, which aids in the metabolic amelioration of the HFD. Furthermore, the authors nicely show that the metabolic phenotype seen with the loss of Fbxw7 can be reverted by silencing of S6K1. In conclusion, Yu and colleagues nicely demonstrate the important role of Fbxw7 within the adipose tissue of mice. The manuscript could be improved further through simple editing of the data, as well as some minor additional experiments. The fact that Fbxw7 binds to multiple proteins within the cell, as shown by the IP (53 binding partners), shows the complexity of its role as part of an E3-ligase. The major question is whether Fbxw7 only binds to S6K1 and if this alone results in the metabolic effects seen in this study, or if Fbxw7 also binds other important proteins, e.g. upstream of S6K1 with these effects only converging down on S6K1. For example, the authors already stated that Fbxw7 is a well-established tumor suppressor gene, which has also been shown to control adipocytes differentiation (PMID: 20534483). In the end, there is very little direct evidence that would support a role of Fbxw7 for brown fat expansion, non-shivering thermogenesis, and energy metabolism in the animal experiments. They are poorly controlled and the HFD feeding results in varying degrees of weight gain. This part of the manuscript needs major revisions.

Thank you for the reviewer's comment. We agree with the reviewer that FBXW7, as an E3 ubiquitin protein ligase, can possibly bind to multiple target genes as shown in our IP-MS results. Of note, our LC-MS/MS results revealed that translation and ribosome-related events were the most significantly affected pathway in in adipose tissues of BAT of FBXW7-FKO mice. This pointed to possible regulation of mTOR signaling, which is highly correlated with increased ribosome protein synthesis and tissue expansion. Consistent with this, overlap of LC-MS/MS and IP-MS data suggested that S6K1, a key protein in mTOR signaling, is a potential FBXW7 substrate, which attracted our attention. Interestingly, further comparison of various key mTOR signaling proteins, including mTOR, Rictor, Raptor and S6K1, in the proteomics pointed out that S6K1 was uniquely upregulated in BAT of FBXW7-FKO mice, suggesting that the metabolic effect observed in this article was mediated by S6K1 rather than the upstream proteins of S6K1.

In addition, it has been reported that FBXW7 controlled adipocyte differentiation by targeting C/EBPalpha for degradation as mentioned by the reviewer (Bengoechea-Alonso MT, et al., 2010). While in our present study, FBXW7 was specifically overexpressed or knocked out with the adiponectin promoter-driven Cre. In the fat tissues of mice, adiponectin only expressed in mature adipocytes but not in preadipocytes. Therefore, in our system, FBXW7 was altered only in mature adipocytes, which may exempt its impact on preadipocytes and adipogenesis. Consistently, we subsequently revealed the glycolytic product lactate from mature adipocytes may affect preadipocytes proliferation in BAT of mice with FBXW7 modulation.

However, though we demonstrated that S6K1 was at least the direct target and central node for metabolic effects of FBXW7 modulation in BAT of mice. we can't fully exclude the possibility that other factors may also be targeted by FBXW7 and affected S6K1 activity. Thus, we have added the limitation of study in the discussion section.

Besides, as per the reviewer's comment, in order to provide more direct evidence to support the role of FBXW7 for brown fat expansion, non-shivering thermogenesis,

and energy metabolism in the animal experiments. We have additionally performed *in vivo* metabolic analysis of FBXW7 specific knockdown in brown adipose tissue by AAV-mediated shRNA delivery. This is achieved by an adeno-associated virus carrying the adiponectin promoter-driven shFBXW7 or control injected locally in BAT.

The results showed that, similar to FBXW7-FKO mice, FBXW7 knockdown in BAT resulted in increased energy expenditure upon CL-316,243 treatment and enhanced thermogenic capacity after cold exposure without changes in body weights or body composition under chow diet (Fig. EV9A-D). Detailed analysis showed that FBXW7 knockdown in BAT of mice caused similar phenotypes as FBXW7 KO mice, and resulted in significantly larger BAT compared to control mice (Fig. EV9E), while lipid contents and number of nuclei per area were similar in both groups (Fig. EV9F, G). Importantly, we found that DNA content per depot was significantly increased in BAT with FBXW7 knockdown, collectively suggesting the increased BAT weights were mainly due to increased cell numbers (Fig. EV9H). Consistent with enhanced energy expenditure, FBXW7 knockdown in BAT caused increased *Ucp1* expression compared to control group (Fig. EV9I, J). These results revealed that FBXW7 knockdown in BAT of mice promoted BAT expansion with enhanced energy expenditure and thermogenesis, as those in FBXW7-FKO mice.

Figure EV9

For other comments, we have made substantial major revision and point-to-point responses to the reviewer's suggestions below.

References:

Bengoechea-Alonso MT, Ericsson J. The ubiquitin ligase Fbxw7 controls adipocyte differentiation by targeting C/EBPalpha for degradation. *Proc Natl Acad Sci U S A*. 2010;107(26):11817-11822.

Major points:

1. Body temperature measurement is not a good indicator of BAT activity, as it includes confounding changes in shivering activity and other cardiovascular physiological mechanisms. Please redo this experiment by subjecting the mice to a

β 3-agonist such as CL316 243 and measure O₂ consumption as described in PMID: 21177944.

Thank you for the reviewer's comment. As per the reviewer's suggestion, we have housed mice at 30°C for a week and subjected mice to 1mg/kg CL-316,243 i.p. injection and measured O₂ consumption as described previously (Cannon et al., PMID:21177944). Though there were no differences in oxygen consumption under thermoneutrality, FBXW7-FKO mice exhibited higher, while FBXW7-FTG mice exhibited lower oxygen consumption upon CL-316,243 challenge, compared to their littermate controls (Fig. EV3K, EV7K).

References:

Cannon B, Nedergaard J. Nonshivering thermogenesis and its adequate measurement in metabolic studies. *J Exp Biol.* 2011;214(Pt 2):242-253. doi:10.1242/jeb.050989

2. For investigating brown fat expansion (what precisely do the authors mean?), a systematic analysis of non-shivering thermogenesis in the animals is warranted.

Thank you for the reviewer's comment. The brown fat expansion refers to adipose tissue hyperplasia due to enhanced preadipocyte proliferation, which results in larger BAT mass (tissue weight) and increased adipocytes numbers (DNA content per fat depot), without changes in adipocyte sizes (number of nuclei per area). We found that FKO mice featured enhanced brown fat expansion under both high fat diet (Fig. 2) and chow diet regiments (Fig. EV8).

In addition to our previous data of energy expenditure, cold tolerance, BAT weights and morphology, numbers of nuclei per area, DNA content per depot, etc., for

a systematic analysis of non-shivering thermogenesis, we further examined energy expenditure upon CL-316,243 i.p. injection and thermogenic capacity with core temperature and thermal imaging of mice under cold exposure in FBXW7-FKO and FBXW7-FTG mice in chow diet regiment, compared to their littermates. The results showed that though there were no differences in oxygen consumption under thermoneutrality, FBXW7-FKO mice exhibited higher, while FBXW7-FTG mice exhibited lower oxygen consumption upon CL-316,243 administration, compared to their separate littermate controls (Fig. EV3K, EV7K). Besides, FBXW7-FKO mice showed enhanced, while FBXW7-FTG mice showed reduced thermogenic capacity under cold exposure, compared to their separate littermate controls (Fig. EV3L, EV7L), suggesting that non-shivering thermogenesis was altered in mice with FBXW7 modulation.

Figure EV3

Figure EV7

3. Please use ANCOVA for determining if the changes in body weight are results of energy expenditure as described in PMID: 22205519.

Thank you for the reviewer's comment. As per the reviewer's request, ANCOVA with body weight as covariant was used to re-analyze the energy expenditure of mice in Figure 2D, 3D, 7D, EV3I and EV7I. The results showed that

the energy expenditure changes of mice are independent of their body weight. The information has been updated in the methods and corresponding figure legends.

4. How many mice were used for the experiments in all the figures? Were these littermate controls? When comparing Figures 3C and 7C, the curves of HFD FKO very much resemble each other. Was this a separate mouse experiment or were the previous data from the experiment in Figure 3 reused? It also appears odd that the authors would choose a different Y-axis labelling for Figure 7C. Please provide all details on statistics in the mouse experiments like repetitions, reproduction, mouse numbers.

Thank you for the reviewer's comment. As per the reviewer's request, we have updated the number of mice in the figure legend for each animal experiment (n=5 for metabolic chamber and n=6 for other experiments throughout the manuscript) in the revised manuscript.

We used littermate controls in our animal experiments. Figure 3C and 7C were independent experiments. Separate mouse experiments were performed and used for analysis for Figure 3C and 7C in our *in vivo* analysis. The growth curves happen to look similar but they are not identical and are calculated from different mice.

The choice of Y-axis is due to the artistic perspective. As per the reviewer's comment, we have used consistent Y-axis for both Figure 3C and Figure 7C. In addition, we have added the repetitions, reproduction, mouse numbers of the mouse experiments in methods section and figure legends of the revised manuscript.

5. It would be valuable to see western blots of mouse tissues, showing that overexpression and knockout of Fbxw7 were achieved on protein level. Similarly, I would like to see that the changes in mRNA expression of Ucp1 in the mice actually show the same effects on protein level.

Thank you for the reviewer's comment. As per the reviewer's request, we have added western blots of FBXW7 in multiple tissues of FBXW7-FTG and FKO mice. The results showed that FBXW7 protein was significantly overexpressed in adipose

tissues, including BAT, iWAT and eWAT, from FBXW7-FTG mice compared to WT littermate controls, while with no change in the liver and muscle (Fig. EV2C-E). Meanwhile, compared with WT littermate controls, FBXW7 was efficiently deleted in three adipose tissues, but not in liver and muscle, of FBXW7-FKO mice (Fig. EV6C-E), indicating that the mouse model of FBXW7 fat-specific overexpression or knockout were successfully established.

Figure EV2

Figure EV6

Moreover, for Fig. 2K, 2M; Fig. 3J; Fig. EV4D, EV4H; EV5M, EV5N; EV8D, EV8I and EV10N, we have added UCP1 protein levels to match UCP1 mRNA data, which showed that UCP1 mRNA and protein levels are consistently changed.

Figure 2

Figure 3

Figure EV4

Figure EV5

Figure EV8

Figure EV10

6. Please always include all data points in the graphs, not just the bar with SD.

Thank you for the reviewer's comment. We have added all data points in the graphs.

Minor points:

7. Figure 2 and 3 E: What's the temperature that the mice we subjected to here? 22°C? 4°C?

Thank you for the reviewer's comment. Mice were subjected at 4°C in the Figure 2E and 3E. We have added the temperature information in the legends of Figure 2E and 3E.

8. Please include proper positive controls in all validation experiments that utilize inhibitors (e.g. for proteasome inhibition with MG-132, show increased ubiquitination)

Thank you for the reviewer's comment. As per the reviewer's request, we have added proper controls in validation experiments.

For Fig. 5D, since MLN4924 has been reported as an inhibitor of NEDD8-activating enzyme (Blank JL, et al., 2013), we have added the positive control showing that MLN4924 treatment effectively inhibited NEDD8 as previously reported and inhibited S6K1 degradation in a dose-dependent manner (Fig. 5D).

For Fig. 5F, we have added cell lysates treated with (Lane 2) or without MG132 (Lane 1). It is clearly shown that MG132 treatment increased ubiquitination, which serves as a positive control. Thus, our data showed that FBXW7 mediated the covalent binding of ubiquitin on S6K1 protein substrates and targeted it for proteasome degradation (Lane 3 in Fig. 5F).

Figure 5

References:

Blank JL, Liu XJ, Cosmopoulos K, et al. Novel DNA damage checkpoints mediating cell death induced by the NEDD8-activating enzyme inhibitor MLN4924. *Cancer Res.* 2013;73(1):225-234. doi:10.1158/0008-5472.CAN-12-1729

9. Please attach all uncropped western blot pictures in the supplement.

Thank you for the reviewer's comment. We have uploaded all uncropped western blot pictures in the supplement files.

10. The manuscript requires a grammar and spell check.

Thank you for the reviewer's comment. We have consulted with native speakers to improve our languages in the manuscript.

11. What about downstream targets of S6K1? For example S6RP? Is it also affected by the change in S6K1 protein levels? An additional Western blot here would be necessary. Furthermore, how is the insulin-signaling pathway influenced by overexpression and KO of Fbxw7?

Thank you for the reviewer's comment. As per the reviewer's request, we have examined phosphorylated RPS6 (p-S6) protein levels, and found that p-S6 levels were increased in BAT of FBXW7-FKO mice, indicating increased protein synthesis of BAT (Fig. EV11B).

Figure EV11

Furthermore, it has been reported that S6K1 phosphorylated IRS1 at S307 site to inhibit the binding of IRS1 to PI3K, which leads to impaired insulin signaling (Um SH, et al., 2004). Indeed, our results showed that p-IRS1 (S307) levels were reduced in BAT and iWAT of FBXW7-FTG mice, while increased in BAT and iWAT of

FBXW7-FKO mice, overall suggesting that S6K1 downstream targets were changed accordingly (Fig. EV11G).

Figure EV11

References

Um SH, Frigerio F, Watanabe M, et al. Absence of S6K1 protects against age- and diet-induced obesity while enhancing insulin sensitivity. *Nature*. 2004;431(7005):200-205. doi:10.1038/nature02866.

12. Figures EV8I-M do not exist. EV20G also does not exist. Please review correct cross-referencing of your figures in the text.

Thank you for the reviewer's comment. We have corrected the mistakes and carefully review the cross-referencing of figures in the manuscript.

13. Why are the ubiquitination and IP assays performed in HEK293T cells and not in brown adipocytes or another adipocyte cell line? The claimed effect of the authors appears to be in adipocytes.

Thank you for the reviewer's comment. We have investigated the ubiquitination and IP assays with ectopic overexpression of Flag-FBXW7 and HA-S6K1 in HEK293T cells due to high transfection efficiency. To further verify the endogenous physical interaction of FBXW7 and S6K1 and the consequent ubiquitination, we have further performed IP assays (Fig. EV11H) and endogenous ubiquitination (Fig. EV11J) in brown adipocytes following the reviewer's suggestion. The results showed that FBXW7 interacted with S6K1 and increased its ubiquitination in brown adipocytes. These data have been added as Fig. EV11H and EV11J.

Figure EV11

14. How does the fat mass of HFD FKO and HFD FKO + shS6K1 change? Only the lean mass is shown. Although figure S12 H shows iWAT and eWAT weight, it does not show how the weights developed throughout the weeks of HFD feeding.

Thank you for the reviewer's comment. HFD FBXW7-FKO + shS6K1 mice showed increased fat mass compared to HFD FKO mice (Figure 7C).

Figure 7

15. Fbxw7 KO leads to increased proliferation through increased lactate production. Why don't the authors prove this by inhibiting the lactate transporter?

Thank you for the reviewer's comment. As per the reviewer's request, we have performed the lactate transporter inhibition experiments. Brown preadipocytes proliferation were increased when treated with culture medium from mature brown adipocytes of FBXW7-FKO mice compared to that from WT mice, which were blocked by lactate transporter inhibitor AZD3965, as shown by EdU staining, CCK8 assay and Ki67 levels (Fig. EV12G-I). In addition, lactate levels changed accordingly in culture medium from mature adipocytes (Fig. EV12J), suggesting that FBXW7 deficiency in brown adipocytes led to increased brown preadipocyte proliferation via enhanced lactate secretion. These data have been updated as Fig. EV12G-J.

Figure EV12

16. One of the targets of Fbxw7 according to the IP data is Pyruvate kinase PKM. If Fbxw7 normally degrades PKM via the UPS, knocking it down could be part of the reason why an increased glycolysis and lactate production was seen.

Thank you for the reviewer's comment. Interestingly, although IP-MS showed that PKM was also a possible target of FBXW7, we found that the transcription levels of glycolytic genes, including Hk2, Pfkfb3, Pfkfb1 and Pfkfb2, were concomitantly increased in BAT and primary brown adipocytes (Fig. 6C, EV12A) from FBXW7-FKO mice, suggesting the glycolysis pathway was enhanced generally on the transcriptional level due to FBXW7 deficiency and S6K1 induction in BAT. It has been reported that mTOR signaling and S6K1 regulate glycolysis (Fan, et al., 2021; Gu, et al., 2015). For example, mTOR signaling and S6K1 are both shown to regulate the transcription factor hypoxia-inducible factor 1 alpha (HIF-1 α) for glycolytic reprogramming, including Pkm2 transcription (Sun Q et al., 2011; Tandon P, et al, 2011; Lee HJ., et al, 2019). Indeed, we showed that S6K1 siRNA or S6K1 specific inhibitor PF4708671 treatment blocked FBXW7 deficiency-induced glycolysis (Fig. EV12O).

However, we agree with the reviewer that in addition to S6K1, other glycolytic genes, such as PKM2 may also be directly bound and regulated by FBXW7, which

may ultimately influence glycolysis in BAT. We have added this information in discussion section.

Figure EV12

Figure EV12

Reference :

Fan. H, Wu. Y, Yu. S, et al. Critical role of mTOR in regulating aerobic glycolysis in carcinogenesis (Review). *Int J Oncol.* 2021;58(1):9-19.

Gu. L, Xie. L, Zuo. C, et al. Targeting mTOR/p70S6K/glycolysis signaling pathway restores glucocorticoid sensitivity to 4E-BP1 null Burkitt Lymphoma. *BMC Cancer.* 2015;15:529.

Sun Q, Chen X, Ma J, et al. Mammalian target of rapamycin up-regulation of pyruvate kinase isoenzyme type M2 is critical for aerobic glycolysis and tumor growth. *Proc Natl Acad Sci U S A.* 2011;108(10):4129-4134.

Tandon P, Gallo CA, Khatri S, Barger JF, Yepiskoposyan H, Plas DR. Requirement for ribosomal protein S6 kinase 1 to mediate glycolysis and apoptosis resistance

induced by Pten deficiency. Proc Natl Acad Sci U S A. 2011;108(6):2361-2365.
doi:10.1073/pnas.1013629108

Lee HJ, Jung YH, Choi GE, et al. O-cyclic phytosphingosine-1-phosphate stimulates HIF1 α -dependent glycolytic reprogramming to enhance the therapeutic potential of mesenchymal stem cells. Cell Death Dis. 2019;10(8):590.
doi:10.1038/s41419-019-1823-7

Referee #2:

The manuscript used human samples and different metabolic disease models and found that FBXW7 is potentially involved in disease progression. Thus, they demonstrated this via FBXW7 overexpression and knockout in the adipose tissue via various experiments. Finally, they pinpointed S6K1 as the substrate of FBXW7 via proteomics. This work is of novelty and physiological significance. However, there are a few concerns to be clarified and to make the work more convincing. Also, the author should correct writing mistakes carefully to make it more professional.

Thank you for the reviewer's comment. We have responded the reviewer's questions point-to-point and thoroughly improved our languages by consulting with native speakers.

Major points

1. The authors used the FBXW7 antibody to examine its levels in different adipose depots of different models. As the work used FBXW7 TG and KO models, however, they have not verified the antibody in those models. To demonstrate regulation of FBXW7 during disease progression, the author should measure FBXW7 protein levels in different adipose depots and liver (corresponding to transcript levels in Fig1A). Moreover, FBXW7 mRNA levels have to be examined in HFD/Ob/Db models (as in Fig1C, 1D), which can provide more information about how FBXW7 is regulated.

Thank you for the reviewer's comment. As per the reviewer's request, the western blots were performed to assess FBXW7 protein levels in multiple tissues of FBXW7-FTG and FKO mice. The results showed that FBXW7 protein was significantly overexpressed in adipose tissues, including BAT, iWAT and eWAT, from FBXW7-FTG mice compared to WT littermate controls, with no changes in the liver and muscle (Fig. EV2C-E). Meanwhile, compared with WT littermate controls, FBXW7 was significantly deleted in three adipose tissues, but not in liver and muscle, of FBXW7-FKO mice (Fig. EV6C-E), indicating that the mouse model of FBXW7 fat-specific overexpression or knockout were successfully established.

Figure EV2

Figure EV6

As per the reviewer's request, we have examined protein levels of FBXW7 in different adipose tissues, livers and pancreas. The results showed that FBXW7 was highly expressed in adipose tissues, which were consistent with transcript levels of *Fbxw7* in Fig. 1A (Fig. EV1A).

Figure EV1

Besides, we examined mRNA levels of Fbxw7 in BAT from HFD, ob/ob, db/db mice and their controls, as well as in human SAT of control and obese patients. The results showed that Fbxw7 mRNA levels were increased in BAT from obese mice and SAT from obese patients, which were consistent with their protein levels as shown in Fig .1C and 1D. These data suggested that Fbxw7 was regulated transcriptionally in thermogenic fat tissues of obese mice and human.

Figure EV1

Figure EV1

2. What is dominant mechanism to mediate the effect of FBXW7 upregulation? cell autonomous suppression of UCP1 in brown and beige adipocytes or suppressed expansion? Are both mechanisms mediated by S6K1 regulation?

Thank you for the reviewer's comment.

In FBXW7-FTG mice, we found that FBXW7 upregulation suppressed BAT expansion, as shown by smaller BAT mass and decreased genomic DNA content per depot. This may be caused by the regulation of glycolysis and lactate production via S6K1. Indeed, we found that the lactate levels in BAT of FBXW7-FTG mice were decreased, and the culture medium from mature primary brown adipocytes of FBXW7-FTG mice had lower lactate secretion (Fig. EV12C, EV12N). Moreover, we found that preadipocyte proliferation was decreased when treated with culture medium from mature brown adipocytes of FBXW7-FTG mice compared to WT mice, as shown by EdU staining, CCK8 assay and Ki67 levels (Fig. EV12K-M).

Furthermore, both UCP1 mRNA and protein levels were decreased in BAT of FBXW7-FTG mice (Fig. EV4D). It has been shown that mTOR signaling increased Pgc1a and Ucp1 gene transcription both in adipocytes and muscles (Xu Y, et al., 2021; Winther S, et al., 2018). Besides, glycolysis likely supports thermogenesis by providing pyruvate and other intermediates to fuel the TCA cycle and oxidative phosphorylation, which may increase Ucp1. Thus, in order to understand whether S6K1 is also responsible Ucp1 levels in mature brown adipocytes, we treated mature brown adipocytes from WT and FBXW-FKO mice with S6K1 inhibitor PF4708671 and found that S6K1 inhibition reduced Ucp1 levels and blocked FBXW7-deficiency induced Ucp1 induction in brown adipocytes (Fig. EV13M).

Overall, these data suggested that both brown fat expansion and Ucp1 levels were regulated by FBXW7-S6K1 axis, which may both contributed to the effects of FBXW7 upregulation on energy expenditure and thermogenesis.

These data have been added in Figure EV12 and EV13M.

Figure EV12

Figure EV12

Figure EV13

Referenc:

Xu Y, Shi T, Cui X, et al. Asparagine reinforces mTORC1 signaling to boost thermogenesis and glycolysis in adipose tissues. *EMBO J.* 2021;40(24):e108069. doi:10.15252/embj.2021108069

Winther S, Isidor MS, Basse AL, et al. Restricting glycolysis impairs brown adipocyte glucose and oxygen consumption. *Am J Physiol Endocrinol Metab.* 2018;314(3):E214-E223. doi:10.1152/ajpendo.00218.2017

3. Can the author provide BMI index of Fig1E, as this might reflect obesity in better way?

Thank you for the reviewer's comment. BMI index of Fig1E has been included in original EV Table S1. BMI index did not differ among three groups with significance by ANOVA analysis, while subjects with GG allele carriers (24.76 ± 5.31) characterized a trend of increased BMI index compared to AA (24.67 ± 2.41) and GG (24.66 ± 1.02) carriers, suggesting fat mass parameter is sensitive to human FBXW7 SNP.

4. Please re-analyse energy expenditure of all the related results in ANCOVA method as this is commonly recommended method for energy expenditure study.

Thank you for the reviewer's comment. As per the reviewer's request, ANCOVA was used to re-analyze the energy expenditure of mice in Figure 2D, 3D, 7D, EV3I and EV7I. The results showed that the energy expenditure changes of mice are independent of their body weight. The information has been updated in the methods and corresponding figure legends.

5. It will make the thermogenesis study more convincing if the authors evaluate the related genes at protein levels, at least for UCP1. In the present version, only mRNA levels are provided.

Thank you for the reviewer's comment. As per the reviewer's request, we further examined UCP1 protein levels in BAT and iWAT from FBXW7-FKO and FBXW7-FTG mice under both chow diet and high fat diet. Indeed, UCP1 levels were significantly increased in BAT and iWAT of FBXW7-FKO mice, while decreased in BAT and iWAT of FBXW7-FTG mice both under NCD and HFD (Fig. 2K, 2M; Fig.

3J; Fig. EV4D, EV4H; EV5M, EV5N; EV8D, EV8I; EV10N). Consistent with increased UCP1 levels in BAT and iWAT of FBXW7-FKO mice, primary brown and beige adipocytes of FBXW7-FKO mice both showed increased UCP1 expression compared to WT (Fig. EV8D; EV8I), while FBXW7 overexpression via adenoviral delivery in beige fat, brown and beige adipocytes also decreased UCP1 levels, which were consistent with their mRNA levels (Fig. 2M; EV5N).

Figure 2

Figure 3

Figure EV4

Figure EV5

Figure EV8

Figure EV10

6. Can the authors explain why the glucose and insulin sensitivity remained largely the same in the FBXW7 KO mice in the HFD feeding conditions (Fig 3D, 3E), considering that the body weight gain is significantly reduced after FBXW7 depletion?

Thank you for the reviewer's comment. In order to understand the reason why the systematic glucose and insulin sensitivity remained the same in the FBXW7-FKO mice under HFD, we performed western blot on insulin signaling pathway on BAT, iWAT, liver and muscle. It has been reported that S6K1 phosphorylated IRS1 at S307 site to inhibit its binding to PI3K, leading to impaired insulin signaling (Um SH, et al., 2004). Consistently, FBXW7-FKO mice showed impaired insulin signaling as shown by decreased p-PI3K, p-AKT and p-GSK3 β levels in BAT and iWAT. However, the insulin signaling of the liver and muscle was improved as shown by increased p-PI3K, p-AKT and p-GSK3 β levels, possibly due to reduced body weights (Fig. EV10I). The combined effects of insulin sensitivity in these tissues may explain the unaltered whole-body glucose metabolism in WT and FBXW7 FKO mice.

These data have been added in Figure EV10I.

Figure EV10

References

Um SH, Frigerio F, Watanabe M, et al. Absence of S6K1 protects against age- and diet-induced obesity while enhancing insulin sensitivity. *Nature*. 2004;431(7005):200-205. doi:10.1038/nature02866

7. According to the results, FBXW7 overexpression in the brown or beige adipocytes should decrease S6K1 and the lactate release to reduce preadipocyte expansion. The authors should measure lactate levels in both the *in vitro* and *in vivo* models after FBXW7 overexpression, just like FBXW7 KO, since FBXW7 upregulation is claimed by the authors to aggravate obesity.

Thank you for the reviewer's comment. Indeed, we have shown in original data Fig. 4G and EV10D that S6K1 was inhibited in BAT and iWAT of FBXW7-FTG mice. As per the reviewer's comment, we have also measured lactate levels in BAT and the culture medium from mature brown adipocytes of FBXW7-FTG mice. The results showed that FBXW7 overexpression decreased lactate secretion both *in vivo* and *in vitro* (Fig. EV12C, N).

8. Please carefully describe in the method part how the authors knock down S6K1 in the iBAT and iWAT together via AAV, including the size of the promoter, the injection method as this might be technically challenging to achieve it.

Thank you for the reviewer's comment. As per the reviewer's request, we have added detailed experimental methods of AAV-GFP or AAV-shS6K1 administration to knockdown S6K1 in BAT and iWAT in materials and methods (Fig. EV14A).

In detail, the adeno-associated virus carrying adipoQ promoter driven shS6K1 or control was injected locally in both BAT and iWAT at the concentration of 2×10^{10} vg (vector genome) in a final volume of 50 μ L to ensure direct action on both thermogenic fats. For BAT, the surgery was performed by making a small incision on

the back of each mouse's scapula area under anesthesia to obtain a clearer view and injected BAT bilaterally with 50 μ L AAV CON or AAV-shS6K1. For iWAT, the skin was disinfected and bilateral inguinal fats were injected subcutaneously with 50 μ L AAV-CON or AAV-shS6K1.

Figure EV14

A

9. Could the author include one group with only S6K1 knockdown in the adipose tissue?

Thank you for the reviewer's comment. As per the reviewer's request, we have added S6K1 knockdown group with injection of AAV-CON or AAV-shS6K1 in both BAT and iWAT of control mice. The results showed that local administration of AAV-shS6K1 increased body weights and fat mass, decreased energy expenditure and impaired thermogenic capacity after cold exposure without changes in locomotor activity and food intake (Fig. EV14A-G). Detailed analysis showed that, similar to FTG mice under HFD, HFD fed WT mice with fat-S6K1 knockdown showed significantly enlarged BAT depot with increased brown adipocyte sizes, more lipid deposition, while decreased numbers of nuclei per unit area (Fig. EV14H-J), genomic DNA content per depot (Fig. EV14J) and Ucp1 mRNA and protein levels (Fig. EV14K), compared to those of WT mice.

Meanwhile, the weights of iWAT and eWAT of HFD mice with fat-S6K1 knockdown were enhanced with increased adipocyte sizes (Fig. EV14L-N), decreased numbers of nuclei per unit area and genomic DNA content per depot, as well as reduced Ucp1 mRNA and protein levels in iWAT, compared to those of HFD WT mice (Fig. EV14O, P). In addition, HFD fat-S6K1 knockdown mice showed increased liver weights and lipid infiltration, as well as elevated serum lipid profiles compared

to WT mice (Fig. EV14Q-T). Taken together, these data suggested that both brown fat expansion and Ucp1 levels were regulated by S6K1.

Figure EV14

Minor points:

1. Fig1C,1D, FigS1A, B, the authors mis-spelled relative as relative. FigS3D, S7D, S9D, Percentage instead of percent might be better.

Thank you for the reviewer's comment. We have amended spelling mistakes in Fig 1C, 1D, Fig EV1A, B and replaced percent with percentage in Fig EV3D, EV7D, EV9D.

2. Thermogenic fat, including brown (BAT) and beige adipose tissues generate heat by decoupling proton gradient from oxidative phosphorylation and release it through uncoupling protein (UCP1) in mitochondrial membrane. Release should be 'releasing'.

Thank you for the reviewer's comment. We have changed "release" to "releasing" in the manuscript.

3. thus are tightly involved in the adipose tissue functionality and onset of obesity. should be, Thus, they are tightly....

Thank you for the reviewer's comment. We have revised "thus are tightly...." to be "Thus, they are tightly...." in the manuscript.

4. To evaluate the functions of FBXW7 in fat biology, we examined Fbxw7 expression levels in various metabolic organs and found that adipose tissues including BAT, inguinal WAT (iWAT) and epididymal WAT (eWAT), should be 'epididymal'.

Thank you for the reviewer's comment. We have changed "epididymal" to "epididymal" in the manuscript.

5. Interestingly, we also observed increased numbers of nuclei per unit area and genomic DNA content per depot, as well as reduced Ucp1 expressions in iWAT of FBXW7-FTG mice. Based on FigS5L, nuclei and genomic DNA should be decreased.

Thank you for the reviewer's comment. We have changed "increased" to "decreased" in the manuscript.

6. these data suggests that FBXW7 aggravates obesity and metabolic dysfunctions in mice, possibly due to impaired tissue expansion and thermogenesis in thermogenic fat. It should be 'suggest'.

Thank you for the reviewer's comment. We have revised "suggests" to "suggest" in the manuscript.

7. The thermogenic capacity of BAT of FBWX7-FTG mice were suppressed as evident by the decreased UCP1 levels (Fig EV4D). This sentence requires rephrasing.

Thank you for the reviewer's comment. We have rephrased this sentence as, 'Meanwhile, the UCP1 levels were decreased in BAT of FBXW7-FTG mice, suggesting impaired thermogenic capacity of BAT of FBXW7-FTG mice (Fig EV4D).

8. Detailed analysis showed that as expected and opposite to FBXW7-FTG's phenotypes, Strikingly, we found that FBXW7-FKO mice had significantly larger BAT compared to WT (Fig EV8A). This sentence requires rephrasing.

Thank you for the reviewer's comment. We have rephrased this sentence as "Detailed analysis showed that in contrast with BAT phenotype of FBXW7-FTG mice, FBXW7-FKO mice had significantly larger BAT compared to WT (Fig. EV8A)."

9. Indeed, recombinant FBXW7 protein mediated S6K1 protein ubiquitination in vitro (Fig 5A) and coimmunoprecipitation assays showed that FBXW7 physically interacted with S6K1 in overexpressed HEK29T and endogenously in BAT (Fig 5B and C). HEK29T should be 'HEK293T'.

Thank you for the reviewer's comment. We have changed HEK29T to "HEK293T" in the manuscript.

10. Since we found that FBXW7 deficiency increased brown fat sizes and functions by impairing adipocyte expansion and thermogenesis, to further understand the

intrinsic mechanisms, we performed flow.... This is a wrong statement according to the results.

Thank you for the reviewer's comment. We have changed "impairing" to "improving" and rephrased this sentence as "Since we found that FBXW7 deficiency increased brown fat sizes and functions by improving adipocyte expansion and thermogenesis, to further understand the intrinsic mechanisms, we performed flow...."

There are so many grammar or wording mistakes as what I pointed out above. I recommend the authors to seek assistance from ones with better language skills.

Thank you for the reviewer's careful reading the valuable comments. We have thoroughly improved our languages by consulting with native speakers.

Referee #3:

This manuscript examined the physiological role of a brown fat abundant E3 ubiquitin protein ligase FBXW7 in thermogenesis and energy homeostasis. Through fat specific overexpression or loss-of-function analysis, the authors concluded that FBXW7 targets S6K1 for ubiquitination and degradation, which induced lactate production. Lactate then act on preadipocytes to promote proliferation and differentiation. Overall, the research is interesting. However, conclusions and molecular mechanisms are needed to be further polished. In addition, the authors need to reformat the manuscript, like Fig EV means figure supplement or? It's hard to follow in the current format. More specifically, I have the following points need to be clarified by the authors.

Thank you for the reviewer's comment. We have prepared point-to-point responses to improve our conclusions and molecular mechanisms following the reviewer's valuable comments.

EV stands for "Extended View", which is required specifically from EMBO Press to replace "Supplementary Information". Besides, we have thoroughly improved our languages by consulting with native speakers.

1. How about brown fat specific FBXW7 KO mice, if the authors conclude the phenotype is due to thermogenesis and BAT manifested a very strong phenotype.

Thank you for the reviewer's comment. We agree with the reviewer that brown fat specific FBXW7-BKO mice may provide more direct evidences to demonstrate the role of FBXW7 in the regulation of BAT physiology and thermogenesis and energy homeostasis. However, due to the limited time to establish a new animal model, we decided to utilize an adeno-associated virus carrying adipoQ promoter driven shFBXW7 or control to inject locally in BAT to examine the phenotypes of FBXW7 specific deficiency in BAT.

The results showed that, similarly with FBXW7-FKO mice, FBXW7 knockdown in BAT resulted in increased energy expenditure upon CL-316,243 treatment and enhanced thermogenic capacity after cold exposure with similar body weights and body composition under chow diet (Fig. EV9A-D). Detailed analysis showed that FBXW7 knockdown in BAT of mice showed significantly larger BAT compared to control mice (Fig. EV9E). Importantly, we found that DNA content per depot was significantly increased in BAT with FBXW7 knockdown, while lipid contents and numbers of nuclei per area were similar in histological analysis (Fig. EV9F-H), suggesting the increased BAT weights were mainly due to increased cell numbers (Fig. EV9H). Consistent with enhanced energy expenditure, FBXW7 knockdown in BAT showed increased Ucp1 expression compared to control group (Fig. EV9I, J). These results revealed that FBXW7 knockdown in BAT of mice cuased BAT expansion for enhanced energy expenditure and thermogenesis, similarly as those in FBXW7-FKO mice.

Figure EV9

2. The authors concluded that FBXW7 expression is negatively correlated with thermogenic fat functionality at the beginning of their story. However, this conclusion is only based on the expression of FBXW7 in brown fat under different conditions such as HFD and in obese animals models (ob/ob and db/db). This conclusion is too farfetched. To consolidate this conclusion, the authors should at least perform a correlation analysis with UCP1 expression.

Thank you for the reviewer's comment. As per the reviewer's request, we have further examined mRNA levels of Fbxw7 and Ucp1 in the HFD, ob/ob and db/db mice models and subcutaneous fat of obese humans. Besides, we performed correlation analysis of Fbxw7 with Ucp1 as the reviewer suggested. The results showed that Fbxw7 was increased in obese mice models and obese human fat (Fig. EV1B, E, H), and was negatively correlated with Ucp1 levels (Fig. EV1C, F, I), suggesting Fbxw7 is involved in the suppression of thermogenic functions and the development of obesity.

Figure EV1

3. Please clarify how the Adipose-specific overexpression of FBXW7 is achieved? With adiponectin-cre, UCP1-cre, PDGFRA-cre or Cre with ERT2? Please check UCP1 expression at protein level. To conclude brown fat function is affected, the authors need to perform respirometry experiment with acute injection of NE or CL. As brown fat is enlarged, it would be necessary to calculate the whole brown fat level of UCP1 both at protein level and mRNA level.

Thank you for the reviewer's comment. FBXW7-LoxP mice were generated by using the EGE system based on CRISPR/Cas9, and CAG-loxP-Stop-loxP-EGE-FBXW7-WPRE-pA was inserted at the Rosa26 site. Then FBXW7 fat-conditional overexpressed mice (FBXW7-FTG mice) was generated by crossing FBXW7-LoxP mice with Adiponectin-Cre mice.

As per the reviewer's request, we further examined UCP1 protein levels in BAT and iWAT from FBXW7-FKO and FBXW7-FTG mice under both chow diet and high fat diet. Indeed, UCP1 levels were significantly increased in BAT and iWAT of FBXW7-FKO mice, while decreased in BAT and iWAT of FBXW7-FTG mice both under NCD and HFD (Fig. 2K, 2M; Fig. 3J; Fig. EV4D, EV4H; EV5M, N; EV8D, EV8I; EV10N). Consistent with increased UCP1 levels in BAT and iWAT of FBXW7-FKO mice, primary brown and beige adipocytes of FBXW7-FKO mice both showed increased UCP1 expression compared to WT (Fig. EV8D; EV8I), while FBXW7 overexpression via adenoviral delivery in beige fat, brown and beige adipocytes also decreased UCP1 levels, which were consistent with their mRNA levels (Fig. 2M; EV5N).

Figure 2

Figure 3

Figure EV4

Figure EV5

Figure EV8

Figure EV10

For evaluation of brown fat function, we further examined energy expenditure with CL-316,243 i.p. injection and thermogenic capacity with core temperature and thermal imaging of mice under cold exposure to systematically analyze the non-shivering thermogenesis of FBXW7-FKO and FBXW7-FTG mice, compared to their littermates. The results showed that though there were no differences in oxygen consumption under thermoneutrality, FBXW7-FKO mice exhibited higher, while FBXW7-FTG mice exhibited lower oxygen consumption when they were i.p injected

with CL-316,243, compared to their separate littermate controls (Fig. EV3K, EV7K). Besides, FBXW7-FKO mice showed enhanced, while FBXW7-FTG mice showed reduced thermogenic capacity under cold exposure, compared to their separate littermate controls (Fig. EV3L, EV7L).

Figure EV3

Figure EV7

As per the reviewer's request, we have calculated the whole brown fat level of UCP1 both at protein and mRNA levels by multiplying protein/mRNA levels per unit by total protein/RNA quantity per BAT. The results showed that FBXW7-FKO mice exhibited significantly increased UCP1 mRNA and protein levels per BAT depot. The methods and results have been updated in the manuscript.

Figure EV8

4. Similar to what were observed in CD mice.... what are CD mice?

Thank you for the reviewer's comment. CD mice mean mice fed with normal chow diet. We have clarified it as "chow diet fed mice" to avoid confusion.

5 please rewrite the sentence "Detailed analysis showed that as expected and opposite to FBXW7-FTG's phenotypes, Strikingly, we found that FBXW7-FKO mice had significantly larger BAT compared to WT (Fig EV8A)."

Thank you for the reviewer's comment. We have rephrased this sentence as "Detailed analysis showed that in contrast with BAT phenotype of FBXW7-FTG mice, FBXW7-FKO mice had significantly larger BAT compared to WT (Fig EV8A)."

6. "Since we found that FBXW7 deficiency increased brown fat sizes and functions by impairing adipocyte expansion and thermogenesis," how could this possible? brown fat function is thermogenesis or?

Thank you for the reviewer's comment. We are sorry for the mistake. We have changed "impairing" to "improving" and rephrased this sentence as "Since we found that FBXW7 deficiency increased brown fat sizes and functions by improving adipocyte expansion and thermogenesis."

7. How does loss of FBXW7 lead to increase of glycolysis or mTOR signaling? Whats the role of S6K1 here? Are glycolysis genes are decreased in FBXW7 overexpressed animals?

Thank you for the reviewer's comment. Our LC-MS/MS results revealed that translation and ribosome-related events were the most significantly affected pathway in in adipose tissues of BAT of FBXW7-FKO mice. This pointed to possible regulation of mTOR signaling, which is highly correlated with increased ribosome protein synthesis and tissue expansion. Consistent with this, overlap of LC-MS/MS and IP-MS data suggested that S6K1, a key protein in mTOR signaling, is a potential FBXW7 substrate, which was then confirmed by a series of molecular analysis in the

present study. Thus, loss of FBXW7 may influence mTOR signaling through dampened degradation of S6K1.

mTOR signaling is vital for glycolysis (Fan, et al., 2021; Gu, et al., 2015). Importantly, it has been shown that S6K1 is critical for hypoxia-inducible factor 1 alpha (HIF-1 α) activation and glycolytic reprogramming (Tandon P, et al, 2011; Lee HJ., et al, 2019). Indeed, preadipocytes proliferation was increased when treated with culture medium from mature brown adipocytes of FBXW7-FKO mice compared to WT mice, which effects were blocked by HIF-1 α inhibitor PX-478, as shown by EdU staining, CCK8 assay and Ki67 levels (Fig. EV12P-S). In addition, lactate levels were changed accordingly, suggesting that FBXW7 KO leads to enhanced glycolytic reprogramming and increased brown preadipocyte proliferation through S6K1/HIF1 α axis.

Figure EV12

Furthermore, we examined glycolytic genes in BAT from FBXW7-FKO mice, the results showed that glycolytic genes were inhibited in BAT from FBXW7-FKO mice, which were in contrast to that in FBXW7-FKO mice.

Figure EV12

References

Fan. H, Wu. Y, Yu. S, et al. Critical role of mTOR in regulating aerobic glycolysis in carcinogenesis (Review). *Int J Oncol.* 2021;58(1):9-19.

Gu. L, Xie. L, Zuo. C, et al. Targeting mTOR/p70S6K/glycolysis signaling pathway restores glucocorticoid sensitivity to 4E-BP1 null Burkitt Lymphoma. *BMC Cancer.* 2015;15:529.

Tandon P, Gallo CA, Khatri S, Barger JF, Yepiskoposyan H, Plas DR. Requirement for ribosomal protein S6 kinase 1 to mediate glycolysis and apoptosis resistance induced by Pten deficiency. *Proc Natl Acad Sci U S A.* 2011;108(6):2361-2365. doi:10.1073/pnas.1013629108

Lee HJ, Jung YH, Choi GE, et al. O-cyclic phytosphingosine-1-phosphate stimulates HIF1 α -dependent glycolytic reprogramming to enhance the therapeutic potential of mesenchymal stem cells. *Cell Death Dis.* 2019;10(8):590. doi:10.1038/s41419-019-1823-7

8. There is no UCP1 protein expression analysis in the whole paper. No brown fat functional analysis like NE or CL test analysis.

Thank you for the reviewer's comment. As per the reviewer's request, we further examined UCP1 protein levels in BAT and iWAT from FBXW7-FKO and FBXW7-FTG mice under both chow diet and high fat diet. Indeed, UCP1 levels were significantly increased in BAT and iWAT of FBXW7-FKO mice, while decreased in

BAT and iWAT of FBXW7-FTG mice both under NCD and HFD (Fig. 2K, 2M; Fig. 3J; Fig. EV4D, EV4H; EV5M, N; EV8D, EV8I; EV10N). Consistent with increased UCP1 levels in BAT and iWAT of FBXW7-FKO mice, primary brown and beige adipocytes of FBXW7-FKO mice both showed increased UCP1 expression compared to WT (Fig. EV8D; EV8I), while FBXW7 overexpression via adenoviral delivery in beige fat, brown and beige adipocytes also decreased UCP1 levels, which were consistent with their mRNA levels (Fig. 2M; EV5N).

Figure 2

Figure 3

Figure EV4

Figure EV5

Figure EV8

Figure EV10

As per the reviewer's request, we further examined energy expenditure with CL-316,243 i.p. injection and thermogenic capacity with core temperature and thermal imaging of mice under cold exposure to systematically analyze brown fat functions of FBXW7-FKO and FBXW7-FTG mice, compared to their littermates. The results showed that though there were no differences in oxygen consumption under thermoneutrality, FBXW7-FKO mice exhibited higher, while FBXW7-FTG mice exhibited lower oxygen consumption when they were i.p injected with CL-316,243, compared to their separate littermate controls (Fig. EV3K, EV7K). Besides, FBXW7-FKO mice showed enhanced, while FBXW7-FTG mice showed reduced thermogenic capacity under cold exposure, compared to their separate littermate controls (Fig. EV3L, EV7L). Overall, these data suggested that FBXW7 impacted brown fat functions in mice.

Figure EV3

Figure EV7

Dear Prof. Ma,

Thank you for submitting your revised manuscript. It has now been seen by all of the original referees.

As you can see, the referees find that the study is significantly improved during revision and recommends publication. However, I need you to address the points below.

- Please address the remaining concerns of the referees #1 and #3.
- We note that FBXW7 blots shown in Fig. EV2C and D are flipped on the horizontal axis. Please clarify.
- Please remove 'Author Contributions' section from the manuscript text.
- As per our format requirements, in the reference list, citations should be listed in alphabetical order and then chronologically, with the authors' surnames and initials inverted; where there are more than 10 authors on a paper, 10 will be listed, followed by 'et al.'. Please see <https://www.embopress.org/page/journal/14693178/authorguide#referencesformat>
- Please fill out and include an author checklist as listed in our online guidelines (<https://www.embopress.org/page/journal/14693178/authorguide>)
- We note that ORCID iD of Dr. Lingyan Xu has not been linked to our manuscript tracking system. EMBO Press policy asks for all corresponding authors to link to their ORCID iDs. You can read about the change under "Authorship Guidelines" in the Guide to Authors here: <https://www.embopress.org/page/journal/14693178/authorguide#authorshipguidelines>

In order to link your ORCID iD to your account in our manuscript tracking system, please do the following:

1. Click the 'Modify Profile' link at the bottom of your homepage in our system.
2. On the next page you will see a box halfway down the page titled ORCID*. Below this box is red text reading 'To Register/Link to ORCID, click here'. Please follow that link: you will be taken to ORCID where you can log in to your account (or create an account if you don't have one)
3. You will then be asked to authorise Wiley to access your ORCID information. Once you have approved the linking, you will be brought back to our manuscript system.

We regret that we cannot do this linking on your behalf for security reasons.

- For technical reasons, our limit for expanded view (EV) figures is 5 (please see our author guidelines <https://www.embopress.org/page/journal/14693178/authorguide#expandedview>). You currently have 14 EV figures. You can either combine figures into 5 expanded view figures, or all or a subset of EV figures could be turned into an appendix file, with the correct nomenclature "Appendix Figure S1" etc. and a table of contents added to the first page. Either way, please update the callouts in the text. Should you opt for the Appendix, please provide a single PDF (nomenclature to name and refer to Appendix items in the main text: Appendix Figure S1, Appendix Table S1, Appendix Supplementary Methods) including a Table of Contents with page numbers on the title page.
- Along similar lines, please submit main figures and EV figures separately as individual production quality figure files (<https://www.embopress.org/page/journal/14693178/authorguide#figureformat>).
- We note that Fig. 3C, Fig. 3J, Table EV2 are currently not called out in the text.
- Table EV2 and EV3 are too large to be EV tables. Thus, please convert them into datasets (whose nomenclature is Dataset EV1 etc.) and update the callouts accordingly.
- All research articles submitted as revised versions must include a structured methods section that includes a Reagents and Tools Table followed by a Methods and Protocols section. Please see <https://www.embopress.org/page/journal/14693178/authorguide#structuredmethods> for further information.
- We note that Table EV4 is a list of oligos used, which should be included in the Reagents and Tools table (https://www.embopress.org/pb%2Dassets/embo-site/Reagents_Tools_Table_TEMPLATE.docx).
- Please change the statement in the 'Data availability' section as 'This study includes no data deposited in external repositories.'
- We note that the study utilizes a publicly available dataset (i.e. microarray genotyping datasets from Shanghai Nicheng Cohort Study (Li et al, 2022)). Dataset needs to be cited separately in the form of data citation (please see <https://www.embopress.org/page/journal/14693178/authorguide#referencesformat>).
- Papers published in EMBO Reports include a 'synopsis' and 'bullet points' to further enhance discoverability. Both are displayed on the html version of the paper and are freely accessible to all readers. The synopsis includes a short standfirst summarizing the study in 1 or 2 sentences (max 35 words) that summarize the paper and are provided by the authors and streamlined by the handling editor. I would therefore ask you to include your synopsis blurb and 3-5 bullet points listing the key experimental findings.
- In addition, please provide an image for the synopsis. This image should provide a rapid overview of the question addressed in the study but still needs to be kept fairly modest since the image size cannot exceed 550 (width) x 300-600 (height) pixels.

Thank you again for giving us to consider your manuscript for EMBO Reports, I look forward to your minor revision.

Kind regards,

Deniz Senyilmaz Tiebe

--

Deniz Senyilmaz Tiebe, PhD
Senior Scientific Editor
EMBO Reports

Referee #1:

The authors have provided a thorough revision.

What remains is that the authors have not (unlike stated) expressed energy expenditure in the ANCOVA using body weight as a covariant.

Referee #2:

The authors addressed all my concerns convincingly I suggest acceptance

Referee #3:

Although significant efforts are being made in revising this manuscript, the authors still used a nonproper way to correct metabolic rate for body mass ($\text{kg}^{0.75}$). The authors should perform the correction according to PMID: 34489606 DOI: 10.1038/s42255-021-00451-2. As energy expenditure or thermogenesis in this manuscript is essential, the paper still need to be revised.

Editor:

- Please address the remaining concerns of the referees #1 and #3.

Thank you for the editor's comment. We have provided point-to-point answers to the referees' questions.

- We note that FBXW7 blots shown in Fig. EV2C and D are flipped on the horizontal axis. Please clarify.

Thank you for the editor's comment. We have re-examined both Figures EV2C and D and the original figures. They were indeed mis-flipped by mistake. The correct figures as Appendix Figure S2C and S2D and original figures as source data were updated.

- Please remove 'Author Contributions' section from the manuscript text.

Thank you for the editor's comment. We have removed the 'Author Contributions' section from the manuscript.

- As per our format requirements, in the reference list, citations should be listed in alphabetical order and then chronologically, with the authors' surnames and initials inverted; where there are more than 10 authors on a paper, 10 will be listed, followed by 'et al.'. Please see

<https://www.embopress.org/page/journal/14693178/authorguide#referencesformat>

Thank you for the editor's comment. we have updated the reference list following the format requirement of the EMBO press.

- Please fill out and include an author checklist as listed in our online guidelines (<https://www.embopress.org/page/journal/14693178/authorguide>)

Thank you for the editor's comment. We have filled out and included an author checklist following the online guidelines.

- We note that ORCID iD of Dr. Lingyan Xu has not been linked to our manuscript tracking system. EMBO Press policy asks for all corresponding authors to link to their ORCID iDs. You can read about the change under "Authorship Guidelines" in the Guide to Authors here:

<https://www.embopress.org/page/journal/14693178/authorguide#authorshipguidelines>

In order to link your ORCID iD to your account in our manuscript tracking system, please do the following:

1. Click the 'Modify Profile' link at the bottom of your homepage in our system.
2. On the next page you will see a box halfway down the page titled ORCID*. Below this box is red text reading 'To Register/Link to ORCID, click here'. Please follow that link: you will be taken to ORCID where you can log in to your account (or create an account if you don't have one)
3. You will then be asked to authorise Wiley to access your ORCID information. Once you have approved the linking, you will be brought back to our manuscript system.

Thank you for the editor's comment. We have linked ORCID iD of Dr. Lingyan Xu (0000-0002-8666-5219) to the manuscript tracking system.

- For technical reasons, our limit for expanded view (EV) figures is 5 (please see our author guidelines

<https://www.embopress.org/page/journal/14693178/authorguide#expandedview>).

You currently have 14 EV figures. You can either combine figures into 5 expanded view figures, or all or a subset of EV figures could be turned into an appendix file, with the correct nomenclature "Appendix Figure S1" etc. and a table of contents added to the first page. Either way, please update the callouts in the text. Should you opt for the Appendix, please provide a single PDF (nomenclature to name and refer to Appendix items in the main text: Appendix Figure S1, Appendix Table S1, Appendix Supplementary Methods) including a Table of Contents with page numbers on the title page.

Thank you for the editor's comment. All EV figures and EV tables have been turned into appendix files, with the correct nomenclature "Appendix Figure S1-S14" and Appendix Table S1, S2 etc.

- Along similar lines, please submit main figures and EV figures separately as individual production quality figure files

(<https://www.embopress.org/page/journal/14693178/authorguide#figureformat>).

Thank you for the editor's comment. Main figures were submitted separately as individual production quality figure files.

- We note that Fig. 3C, Fig. 3J, Table EV2 are currently not called out in the text.

Thank you for the editor's comment. We have called out Fig. 3C, Fig. 3J and Table EV2 (Updated to Dataset EV1) in the manuscript.

- Table EV2 and EV3 are too large to be EV tables. Thus, please convert them into datasets (whose nomenclature is Dataset EV1 etc.) and update the callouts accordingly.

Thank you for the editor's comment. We have converted Table EV2 and EV3 into datasets as Dataset EV1 and EV2, and updated the callouts accordingly in the manuscript.

- All research articles submitted as revised versions must include a structured methods

section that includes a Reagents and Tools Table followed by a Methods and Protocols section. Please see

<https://www.embopress.org/page/journal/14693178/authorguide#structuredmethods>

for further information.

Thank you for the editor's comment. We have updated and submitted a Reagents and Tools Table.

- We note that Table EV4 is a list of oligos used, which should be included in the Reagents and Tools table

(https://www.embopress.org/pb%2Dassets/embo-site/Reagents_Tools_Table_TEMPLATE.docx).

Thank you for the editor's comment. We have added the information of Table EV4 (Updated to Appendix Table S2) to the Reagents and Tools Table.

- Please change the statement in the 'Data availability' section as 'This study includes no data deposited in external repositories.'

Thank you for the editor's comment. We have changed the statement in the 'Data availability' section as 'This study includes no data deposited in external repositories.'

- We note that the study utilizes a publicly available dataset (i.e. microarray genotyping datasets from Shanghai Nicheng Cohort Study (Li et al, 2022)). Dataset needs to be cited separately in the form of data citation (please see <https://www.embopress.org/page/journal/14693178/authorguide#referencesformat>).

Thank you for the editor's comment. The dataset used in this study has been deposited and cited separately as following:

E3 Ligase FBXW7 Regulates Brown Fat Expansion and Browning of White Fat (2024) OMIX of the China National Center for Bioinformation/Beijing Institute of Genomics, Chinese Academy of Sciences OMIX007667

(<https://ngdc.cncb.ac.cn/omix/preview/XofJfw3S>) [DATASET]

• Papers published in EMBO Reports include a 'synopsis' and 'bullet points' to further enhance discoverability. Both are displayed on the html version of the paper and are freely accessible to all readers. The synopsis includes a short standfirst summarizing the study in 1 or 2 sentences (max 35 words) that summarize the paper and are provided by the authors and streamlined by the handling editor. I would therefore ask you to include your synopsis blurb and 3-5 bullet points listing the key experimental findings.

Thank you for the editor's comment. The synopsis blurb is as following:

“FBXW7 regulates brown fat expansion and thermogenesis via targeting S6K1 for ubiquitination and degradation, which impacts glycolysis and brown preadipocyte proliferation via lactate, thus serves as a potential therapeutic target for obesity and metabolic diseases.”

The bullet points are as following as:

- FBXW7 expression is negatively correlated with thermogenic fat functionality.
- Adipose-specific overexpression of FBXW7 exacerbated obesity and metabolic dysfunctions.
- Adipose-specific ablation of FBXW7 induces BAT expansion and browning of white fat, thus protects against obesity in mice.
- FBXW7 negatively regulate ribosomal protein synthesis by targeting S6K1 for ubiquitination and degradation, which in turn impacts glycolysis and brown preadipocyte proliferation via lactate.

• In addition, please provide an image for the synopsis. This image should provide a rapid overview of the question addressed in the study but still needs to be kept fairly modest since the image size cannot exceed 550 (width) x 300-600 (height) pixels.

Thank you for the editor's comment. We have provided an image for the synopsis.

Thank you again for giving us to consider your manuscript for EMBO Reports, I look forward to your minor revision.

Thank you for giving us the great opportunity for revising our work for your prestigious journal.

Referee #1:

The authors have provided a thorough revision.

What remains is that the authors have not (unlike stated) expressed energy expenditure in the ANCOVA using body weight as a covariant.

Thank you for the reviewer's comment. As per the reviewer's suggestion, ANCOVA with body weight as covariant was used to re-analyze the energy expenditure of mice. Data was shown in Appendix Figure S3I, S5B, S7I, S10B, S13C and S14E. The results showed that the FBXW7-mediated energy expenditure changes in mice were independent of their body weights. The information has been updated in the methods and corresponding figure legends.

Appendix Figure S3

Appendix Figure S5

Appendix Figure S7

Appendix Figure S10

Appendix Figure S13

Appendix Figure S14

Referee #2:

The authors addressed all my concerns convincingly I suggest acceptance.

Thank you for the reviewer's valuable comment.

Referee #3:

Although significant efforts are being made in revising this manuscript, the authors still used a nonproper way to correct metabolic rate for body mass (kg 0.75). The authors should perform the correction according to PMID: 34489606 DOI: 10.1038/s42255-021-00451-2. As energy expenditure or thermogenesis in this manuscript is essential, the paper still need to be revised.

Thank you for the reviewer's comment. In the previous revised version, we analyzed respirometry experiment of mice upon CL-316,243 with correcting

metabolic rate for body weight (kg 0.75) as described previously (Golozoubova *et al.*, 2006, PMID:21177944).

As per the reviewer's suggestion, We have re-analyzed respirometry experiment of mice upon CL-316,243 as suggested (Sveidahl *et al.*, Cell, 2021). Similar as previously presented, though there were no differences in oxygen consumption under thermoneutrality, FBXW7-FKO mice exhibited higher, while FBXW7-FTG mice exhibited lower oxygen consumption upon CL-316,243 challenge, compared to their littermate controls (Appendix Figure S3L, S7L and S9C).

Appendix Figure S3

Appendix Figure S7

Appendix Figure S9

Additionally, we further performed ANCOVA analysis with body weight as covariant to re-analyze the energy expenditure of mice in Figure 2D, 3D, 7D, EV3I, EV7I and EV14D as described (Müller *et al.*, 2021, PMID:34489606). The data was updated in Appendix Figure S3I, S5B, S7I, S10B, S13C and S14E. The results showed that the FBXW7 mediated energy expenditure changes of mice were

independent of their body weight. The information has been updated in the methods and corresponding figure legends.

Appendix Figure S3

Appendix Figure S5

Appendix Figure S7

Appendix Figure S10

Appendix Figure S13

Appendix Figure S14

Reference:

Golozoubova V, Cannon B, Nedergaard J. UCP1 is essential for adaptive adrenergic nonshivering thermogenesis. *Am J Physiol Endocrinol Metab.* 2006 Aug;291(2):E350-7.

Sveidahl Johansen O, Ma T, Hansen JB, et al. Lipolysis drives expression of the constitutively active receptor GPR3 to induce adipose thermogenesis. *Cell.* 2021;184(13):3502-3518.e33.

Müller TD, Klingenspor M, Tschöp MH. Revisiting energy expenditure: how to correct mouse metabolic rate for body mass. *Nat Metab.* 2021 Sep;3(9):1134-1136.

Prof. Xinran Ma
East China Normal University
Institute of Biomedical Sciences and School of Life Sciences
Shanghai Key Laboratory of Regulatory Biology
Shanghai 200241
China

Dear Prof. Ma,

Thank you for submitting your revised manuscript. I have now looked at everything and all is fine. Therefore, I am very pleased to accept your manuscript for publication in EMBO Reports.

Before transferring your manuscript to our production, I need your input on one more point. I have made minor changes in the items below. Please take a look and confirm these changes, or feel free to propose further changes. Thank you.

Title:

E3 Ligase FBXW7 Suppresses Brown Fat Expansion and Browning of White Fat

Abstract:

Thermogenic fat, including brown and beige fat, dissipates heat via thermogenesis and enhances energy expenditure. Thus, its activation represents a therapeutic strategy to combat obesity. Here, we demonstrate that levels of F-box and WD repeat domain-containing 7 (FBXW7), an E3 ubiquitin protein ligase, negatively correlate with thermogenic fat functionality. FBXW7 overexpression in fat suppresses energy expenditure and thermogenesis, thus aggravates obesity and metabolic dysfunctions in mice. Conversely, FBXW7 depletion in fat leads to brown fat expansion and browning of white fat, and protects mice from diet induced obesity, hepatic steatosis and hyperlipidemia. Mechanistically, FBXW7 binds to S6K1 and promotes its ubiquitination and proteasomal degradation, which in turn impacts glycolysis and brown preadipocyte proliferation via lactate. Besides, the beneficial metabolic effects of FBXW7 depletion in fat are attenuated by fat-specific knockdown of S6K1 in vivo. In summary, we provide evidence that adipose FBXW7 acts as a major regulator for thermogenic fat biology and energy homeostasis and serves as potential therapeutic target for obesity and metabolic diseases.

Congratulations on a nice work!

Kind regards,

Deniz Senyilmaz Tiebe

--

Deniz Senyilmaz Tiebe, PhD
Senior Scientific Editor
EMBO Reports
